# The relative effectiveness of law enforcement policies aimed at reducing illegal trade: Evidence from laboratory markets

Chian Jones Ritten[1]*, Christopher Bastian[1], Owen Phillips[2]

**1** Department of Agricultural and Applied Economics, University of Wyoming, Laramie, Wyoming, United States of America, **2** Department of Economics, University of Wyoming, Laramie, Wyoming, United States of America

\* chian.jonesritten@uwyo.edu

**Data Availability Statement:** All relevant data are within the manuscript and its Supporting Information files.

## Abstract

Despite recent emphasis and implementation of national and international anti-money laundering policies, illegal product markets, and their associated illicit profit remain a global problem. In addition to law enforcement aimed at reducing money-laundering, enforcement also takes place during (1) the production (e.g. crop eradication) and (2) sale (e.g. seizure of products during transportation that interrupts buyer and seller transactions) of the illegal product. Since funds for enforcement come from limited budgets, understanding where in this production-trade-laundering cycle law enforcement is most impactful becomes a global question. Using laboratory experimental markets and a seizure rate of 20%, we find that law enforcement focused on seizing laundered profits does little to reduce illegal market activity when compared to no law enforcement, suggesting that focusing law enforcement on money laundering will likely be ineffective at reducing crime. Results further show the amount of illicit trade is nearly 32% lower when law enforcement is focused at the point of sale, and there may be additional economic incentives that reduce illicit trade in the long run when compared to no law enforcement. Enforcement at the point of production also reduces market activity, but not as effectively as enforcement at the point of sale. Lastly, the empirical findings deviate from equilibrium predictions, suggesting law enforcement policy based on theory alone may lead to inefficient allocation of limited law enforcement resources.

## Introduction

Illicit trade from outlawed markets is a worldwide problem. Numerous markets outlawed by one or more governing bodies exist globally [1]. These markets often are outlawed due to society viewing the products as repugnant [2]. Broadly characterized, these outlawed markets involve prohibited consumption of goods or services, the unauthorized sale of regulated commodities, the sale of goods that infringe upon intellectual property rights, the sale of goods that do not conform to local standards, goods sold to escape local excise tax and tariffs, and/or the sale of stolen goods [3]. The most common illegal markets involve drugs, the sale of counterfeit

**Funding:** The laboratory experiments were funded through the Paul Lowham Research Fund, University of Wyoming, received by C.B. and O.P. (no grant number or URL are available). The funders had no role in the study design, data collection and analysis, decision to publish, or preparation of the manuscript.

**Competing interests:** The authors declare that no competing interest exist other than the funding through the the Paul Lowham Research Fund, University of Wyoming, which had no role in the study design, data collection and analysis, decision to publish, or preparation of the manuscript.

goods, human trafficking, wildlife poaching, arms and weapons dealing, and transactions for stolen timber, art, gold, human organs, and diamonds [1].

Globalization has provided new opportunities for the proliferation of illegal product distribution. As a result, many international policies exist to halt the production and distribution of illegal products, and the laundering of illicit profit. These policies mainly use the seizure of assets throughout the production-trade-laundering cycle of illegal product markets worldwide [4]. Other penalties, including fines and prison terms, can be conceptualized as a seizure of assets from a broader perspective. Asset seizure and forfeiture is popular because it can reduce trade activity and profits (and therefore punish) of all entities involved, even though it may be difficult to identify all involved parties [5]. In 2020, nearly $600 million worth of illegal drugs and related assets were seized in the United States alone [4].

Although law enforcement polices exist that focus on the production or sale of illegal products, recent emphasis has been placed on heightened enforcement aimed at the laundering of illicit profit. Generally, illegal trading relies on cash transactions, where the cash must be disguised to avoid legal entanglements. This cash is frequently passed through banks and other financial institutions in order to make its source difficult to trace [6]. This form of money laundering disguises the source of proceeds from criminal activity by making it appear as if earnings are legitimate. The amount of cash flowing through illegal transactions is estimated to account for two to five percent of global GDP yearly [6]. In addition to potential societal harm from illegal trade activity, money laundering and the injection of laundered funds into new ventures create economic damage. It is estimated that an increase of USD 1 billion in money laundering can reduce economic growth between 0.03 and 0.06 percentage points [7].

Various other policies focus on other stages throughout the production-trade-laundering cycle. For instance, the Drug Enforcement Administration (DEA) uses crop eradication to reduce the quantity of illegal drugs before they can hit the market. In 2019, the DEA eradicated over 4 million marijuana plants and seized nearly $30 million dollars of cultivator assets [8]. Additionally, other policies include street-level enforcement to interrupt transactions and border enforcement [9] to capture assets during the sale of the illegal products. Border enforcement focuses on seizing illegal products as they cross international boundaries. Since most of these products are in transit to their buyer, seizure at the border, like street-level enforcement, aims law enforcement actions during the sale or trade of illegal products. Although these policies aimed at the other stages in the production-trade-laundering cycle can be impactful, anti-money laundering policies have been the focus of recent anti-crime campaigns.

Numerous national and international anti-money laundering policies and entities have been created over the last 40 years. Examples of such entities include the Financial Action Task Force, the Global Programme Against Money Laundering, and The International Convention Against Transnational Organized Crime, initiated through global organizations such as the United Nations and International Monetary Fund (see [10] for a review). Although increases in anti-money laundering enforcement from these programs is projected to decrease crime [11], many studies find that the effectiveness of these initiatives is limited (e.g. [12]). For instance, Ferwerda et al. [13] find that the Financial Action Task Force's (an independent inter-governmental body that develops and promotes policies to protect the global financial system against money laundering) current method of blacklisting countries for money laundering (used as an attempt to reduce money laundering) may not prevent money laundering, and may further reduce the quality of national statistics on money laundering. Further, Deleanu [12] finds evidence that current policy does not incentive accurate reporting by countries of money laundering, supporting the notion that national statistics on money-laundering activity are not accurate, and may even be misleading.

In another instance of how current anti-money laundering policies may have limited effectiveness, Takáts [14] finds that current polices requiring banking institutions to identify and report suspicion of money laundering may backfire. The Suspicious Activity Report, introduced in 1996 by the Financial Crime Enforcement Network for reporting of any suspicious activity, is one such policy. To incentivize compliance, banks and other financial institutions are charged a fee for not reporting instances of money laundering. Current increases in this fee can cause institutions to over-report potential instances of money laundering, creating a 'cry wolf' phenomenon, which reduces the efficacy of the program to reduce money laundering, and thus crime [14].

This paper focuses on the efficacy of policies against illegal product markets and the laundering of profit from illicit trade. More specifically, we aim to understand if law enforcement against illegal product markets are more effective when focused on the resulting money laundering of illegal profit, or on the criminal activity itself. Much of the answer depends on behavioral issues related to where in the production-trade-laundering cycle the seizure of assets occurs from law enforcement action. Thus, we use laboratory market experiments where there is a risk of seizure at various points along the production-trade-laundering cycle to help understand the effectiveness of law enforcement policy. We find that anti-money laundering policies will likely do little to reduce the level of illegal market activity when compared to no law enforcement, suggesting that focusing law enforcement on money laundering will likely not reduce illicit trade. We further find that law enforcement policies focused at the point of illegal product trade is most effective at combating this type of crime.

## Literature on the economics of money laundering, crime, and illicit trade

Becker's [15] work on crime and punishment begins a formalized way of thinking about the economics of crime (see [16] for a review). Becker's work shows that optimal law enforcement policies against criminal activity are a constrained optimization problem, and the per dollar impact of enforcement options should be equal at the margin. Yet, this is often not achieved in practice [17]. Stigler [18] emphasizes that effective policy needs to take into account the marginal deterrence of law enforcement actions across crimes. The optimal marginal deterrence level (i.e. the change in legal punishment between types of crimes) must be set to incentivize criminals to choose the least socially harmful offense (e.g. if a criminal is executed for a minor offense and murder, there is no marginal deterrence for murder).

Despite a growing literature on optimal law enforcement (see [19] for a review), a paucity of economic analyses regarding how crime is funded exists in the literature. This void is partially due to ignoring what is done with cash, and particularly money laundering, within traditional theoretical models of crime [20]. Masciandaro [20] creates a theoretical framework to understand the mechanics of illegal finance markets. This work assumes money laundering can be understood through an analysis of criminal behavior, where laundering is part of a bundle of illegal actions and is a very close complement to the illegal activity, such that if all laundering were stopped there would be no associated illegal undertakings [20]. Villa et al. [21] expands the economic analysis of illicit trade and money laundering by presenting a theoretical model of long-run growth while incorporating illicit activities. In the model, an improvement in government efficiency against crime increases detections and seizure rates. Specific to money laundering enforcement, Ferwerda [11] empirically finds that policy against laundering has a negative impact on laundering rates, and that international cooperation against laundering has the most robust impact on crime. Such cooperation increases the risk of detection and ultimately punishment. Imanpour et al. [22], using a theoretical model, suggest that optimal law enforcement budget share directed towards money laundering should be near 35%. Yet,

there are many anomalies found in studies of money laundering and crime, suggesting problematic aspects that these theoretical models cannot explain (see [23]).

As a way to circumvent some of these theoretical issues, some economic studies on crime have used field experiments. Levitt [24], through a natural experiment on changes in the relative punishment of juvenile and adult criminals, found criminal behavior is highly responsive to changes in law enforcement punishment. Further Fisman and Miguel [25] find that when enforcement agencies were given authority to penalize parking violators, there was a drastic drop in such violations. Killias et al. [26] find that increasing the probability of detection, and thus punishment in fare dodging on trains in Zurich, decreases crime in a non-linear fashion. While these studies offer insights as to the potential effectiveness of punishment of criminal actions in general, they offer less insight into illegal market activity since both the buyers' and sellers' incentives and behaviors have to be understood to curb illegal trade.

For consumers to demand a product through an illegal market, either there is no legal market for that product, or the price through the illegal market is lower than through legal markets. Further, the buyer must be willing to accept the risk of punishment, which ultimately reduces benefit from the purchased product. A seller's willingness to supply product in an illegal market depends on profitability, which is a function of production costs and risk of punishment [27].

Law enforcement reduces production and consumption of illegal products via raising production costs and reducing consumption benefit through the risk of enforcement activities, such as asset seizures and fines. Specific to illicit trade, Becker at el. [28] present a theoretical model of markets that shows optimal law enforcement expenditure should account for the elasticity of supply and demand for illegal products. Yet in experimental bargaining environments, even accounting fully for the elasticity of demand and supply in the market, outcomes vary considerably from the predicted outcomes of various policies (e.g. [29–32]). For instance, effective taxes, which could be interpreted as fines, have different frames to market agents and unpredictable incidences (e.g. [32–34]). In this light, different forms of asset seizure in illegal markets will likely have impacts that are difficult to predict from theory alone.

Optimal law enforcement research recognizes the need to understand deterrence levels of criminal activity across different possible punishment strategies given budget constraints. Yet, no study compares the effectiveness of multiple law enforcement policies aimed at illegal markets and related money laundering while accounting for the way policies interact with agent behavior, potentially causing enforcement mechanisms to deviate from theory. We address this research gap.

## Methods

Existing data on crime and enforcement is limited, and the quality of that available has been criticized (see [17]). Given the lack of reliable data available for empirical analyses we create an experimental laboratory market environment that mimics the risk of loss from punishment, i.e. asset seizure to both buyers and sellers (producers) trading in illegal product markets. To mimic asset seizure risk, the experimental design asks subjects as producers (sellers) to make decisions when there is some probability of loss along a production-trade-laundering cycle. Subjects as buyers have a risk of loss at the point of sale or in their efforts to launder money. Subjects in our experiment *are not* asked to make decisions as potential criminals, but they make choices where there is a probability of loss, similar to that experienced via law enforcement through asset seizure. Using terms like 'illegal', 'illicit', or 'money laundering', may bring unknown (to the experimenter) impressions or memories that impact behavior of subjects [35]. Our market needs to parallel the important features that may impact market function

without affecting participants' response to the economic incentives in an unpredictable way. Our setup allows us to measure the impact of loss at the different points where asset seizure may occur. Since the impact of various punishments on market outcomes may vary from theory, our approach of using a laboratory experimental market can improve our understanding of how law enforcement resources are best utilized to combat illicit trade.

Many market experiments use a double auction institution, as it is thought to yield results mimicking the competitive equilibrium [36]. However, the market institution through which illegal products reach the consumer is bilateral bargaining. Given this reality, past research has used a bilateral bargaining market for illegal product sales (e.g. [37]). Since our intent is to understand market outcomes in real-world illegal product markets, we use bilateral bargaining, as it offers closer parallelism than the double auction to these markets [37]. We assume that a buyer and seller come together to negotiate a trade after the good is produced. Hence the producer has an inventory of goods when prices are negotiated, and accordingly we model a market delivery system with advanced production (i.e. spot market). As a result, sellers also face inventory loss risk, where they may be left with unsold goods at the end of bargaining and selling.

No law enforcement policy is completely effective at seizing all assets. For instance, illegal crop production may be geographically dispersed, limiting law enforcement's ability to seize all production in one fell swoop. Other illegal goods may also be distributed across distinct geographic markets, forcing law enforcement to undertake numerous independent searches for the goods. Further, illicit profit may be laundered through multiple avenues, limiting the probability that all funds are seized by law enforcement. To mimic this reality, our experimental procedure includes a fixed probability of loss on each individual unit of illegal product during production or trade, or a fixed probability of loss on illicit profit in the experiment.

## Details of the experimental procedure

The use of economic experiments to understand market outcomes is well established (e.g. [29–32,38–41]). If market agents involved in illegal market activity respond to incentives in the same way as those engaged in legal activities (see [42]), market experiments become a reliable source of data. The experimental market methods employed in this study follow standard procedures [35,36].

We construct a simple bargaining environment in the laboratory, which were approved by the University of Wyoming's Institutional Review Board. Four buyers and four sellers negotiate trades for a homogenous product known as a unit. All bargaining activity is carried out on a computer network. Buyers and sellers are randomly paired and given one minute (known as a bargaining round) to reach a price agreement on as many units available. At the end of the minute, a new pairing is made. Buyers and sellers are matched three times to give sellers the opportunity to sell all of their inventory. If inventory is not sold by the end of the third match (or bargaining round) it becomes a sunk cost. A trading period consists of three one-minute bargaining rounds, during which sellers earn profit by agreeing on a trade price that is above their unit cost. Buyers earn a profit by agreeing to trade at a price below their redemption value (the amount they could resell the unit for to the experiment director). Each experimental market session has at least 20 trading periods.

Table 1 describes the experimental unit cost and redemption schedules. The use of predefined supply and demand schedules that are consistent across treatments (Table 1), allows us to test for differences between predicted equilibria and outcomes, and test for differences across treatments involving seizure. Since the objective is to understand market outcomes, individual trade data are aggregated for each session across treatments and include no personal identifying information.

**Table 1. Per-unit buyer redemption values and seller production costs (Tokens[i]).**

| Unit | Buyer Redemption Value | Seller Production Cost |
| --- | --- | --- |
| 1 | 130 | 30 |
| 2 | 120 | 40 |
| 3 | 110 | 50 |
| 4 | 100 | 60 |
| 5 | 90 | 70 |
| 6 | 80 | 80 |
| 7 | 70 | 90 |
| 8 | 60 | 100 |

[i]The artificial currency called "tokens" is exchanged for real dollars: 100 tokens equal one dollar.

To account for the reality that law enforcement is unable to stop all illegal trade, we have set the probability of product or token seizure at 20%. This loss can occur: (1) during production (and thus only the seller is directly impacted), (2) during trade (where both the buyer and seller forfeit the traded good), and (3) during the laundering of illicit profit from trading. At points (1) and (2), sellers and/or buyers have a 20% chance of loss of units to mimic a seizure of units. At point (3), the laundering process occurs when subjects must convert their earnings in a fictitious currency called "tokens" to dollars and cents (each token earned in the experiment has a 20% chance of being seized). We use multiple treatments to measure the efficacy of asset seizure at each one of these points: loss at the point of production, loss at the point of sale, and three money laundering treatments, depending on whether the buyer, seller, or both risk loss when laundering tokens.

We use a between-subjects design where subjects were recruited to participate in only one session and treatment. Subjects selected a session to participate in, after which, the treatment was randomly assigned to each session. Each session had eight participants. Experimental sessions are constructed as follows:

1. Participants in the community are recruited through various outlets (e.g. Facebook, email) to participate in one session of an economic experiment held in a dedicated experimental lab. Given the nature of the experimental design and inability to observe criminal behavior, we use participants from the general population. According to induced-value theory, in an experiment that provides the proper reward and meets the conditions of monotonicity, dominance, and salience, the innate characteristics of subjects becomes irrelevant [35]. Fréchette [43] concludes that experimental outcomes are generally consistent across different subject pools, supporting the use of the general population as subjects. The use of students or the general population is commonly used in the experimental market literature (e.g. [29,31,40,41].

2. Eight subjects (four buyers and four sellers) in a session are randomly assigned to only one treatment. These subjects are given detailed instructions that explain the market and rules of trade for that specific treatment, and provide written consent to participate (see S1 Text for instructions). Eight subjects in a session is common practice in experimental market studies since this number is found to be sufficient to obtain competitive outcomes (see [44]). The instructions provide information on trading rules, how trades are made, how profits are made, and treatment specific information (see S1 Text). Participant questions are answered without coaching.

3. Participants are randomly assigned as either buyers or sellers and remain in this role throughout the experimental session. Sellers are provided with a supply schedule that details unit production costs, and buyers receive a demand schedule that lists unit redemption values (see Table 1). Individuals are only given information on their own cost or redemption schedule and are not given information about the role or schedule of other participants.

4. Subjects participate in at least one practice period (using different unit cost and redemption values than in the actual experiment). Practice periods are continued until session participants indicate they understand the procedures and are ready to move forward with the experiment.

5. Once the practice periods conclude, the experiment begins. At the beginning of a trading period, sellers make a production decision on how many units to produce (up to eight units; see Table 1). Once all sellers have made their production decisions, buyers and sellers are randomly matched to bilaterally negotiate trades of units to earn tokens. Each trading period consists of three one-minute bargaining rounds. In each round, a randomly matched buyer and seller make offers and counteroffers over the sale of a unit. Once the pair successfully negotiates the trade of Unit 1, they begin negotiating over the sale of Unit 2, and so on. At the end of each round, participants are randomly matched again. Sellers can trade as many units as produced for that period. No inventory carryover is allowed between trading periods (three bargaining rounds), so for any unsold units produced, sellers lose the production cost. Buyers can purchase up to eight units (see Table 1). After each trading period, profits and accumulated earnings (in terms of tokens) are posted for each subject privately. Participants take part in at least 20 trading periods. To avoid any strategic behavior in the last trading period [31], we employ a random stop with a 20% chance for a session to stop after the 20th period and an 80% chance for the experiment to continue for another period. Each subsequent period has a 20% chance of being the last period.

6. All subjects are given a beginning balance of $15.00. Participants are paid in cash based on their accumulated earnings over the trading periods in addition to the beginning balance. Total earnings, including the initial balance, for individual participants averaged $34, based on individual performance and the treatment in the experimental sessions. Each experimental session concluded in approximately two hours.

To achieve our research objective, 6 treatments (No Seizure, Seller Profit Seizure, Buyer Profit Seizure, Both Profit Seizure, Product Seizure, and Trade Seizure) are conducted with 6 replications (sessions) each, for a total of 36 experimental sessions with 288 participants. The No Seizure treatment serves as a baseline with no probability of asset seizure anywhere along the production-trade-laundering cycle. All other treatments have a 20% chance of loss at the designated point along the production-trade-laundering cycle.

Each of the three money laundering treatments mimics different laundering needs for market agents. In the three Profit Seizure treatments there is a chance of loss for the respective group when converting tokens to cash. In the Seller Profit Seizure treatment, only the seller must launder illicit profits, and the redemption schedule in Table 1 induces demand, where the buyer does not need to launder profits because they are either the final consumer of the product, or in a situation where money laundering is unnecessary. In the Buyer Profit Seizure treatment, only the buyer, and not the seller, must launder illicit profits. In the Both Profit Seizure treatment, the buyer and seller both must launder earnings. We conduct all three treatments to understand the potential impacts of profit laundering across market agent roles. We recognize there are transaction costs associated with the laundering of illicit profit. When law

enforcement seizes laundered money, for example, criminals must find new avenues to launder their illicit profit, which can be very costly. Our experimental method does not include these costs in order to isolate the impact of law enforcement without confounding impacts from other factors.

In the Product Seizure treatment there is a chance of loss at the production stage. If a unit of production is seized, sellers lose their production costs (Table 1), and are unable to sell the product to a potential buyer. In the Trade Seizure treatment there is a chance of loss at the point of trade. If a unit is seized in this treatment, the buyer and seller lose any negotiated profit from the trade, and the seller loses their production costs.

## Equilibrium predictions

Given our research objective we first examine the predicted equilibria and earnings across treatments to understand the theoretically expected impacts of seizure. Assuming a competitive market, theory predicts that the relative magnitude and elasticity of the supply and demand schedule determine the predicted equilibria and outcomes for buyers and sellers in this market. Each treatment uses the same base supply and demand schedules (Table 1), but based on the specific risk of asset seizure in a given treatment, the effective schedule(s) change, leading to different predicted market equilibria across treatments.

In the No Seizure base treatment, there is no risk of random loss. Given the schedules presented in Table 1, the resulting predicted equilibrium price is 80 tokens per unit (Fig 1). Due to the step function of the supply and demand schedules, the predicted market equilibrium (four buyers and four sellers) quantity is a tunnel between 20 and 24 units (Fig 1; Table 2).

At equilibrium, the earnings of individual buyers and sellers will both be 150 tokens, generating a total market surplus of 1,200 tokens (Table 2).

In the Seller Profit Seizure treatment, seizure of funds occurs at the point where sellers convert tokens to dollars in an effort to launder tokens. There is no risk of loss of units, but each earned token has a 20% probability of being confiscated. Since only sellers face this risk, the effective supply schedule changes, as depicted in Eq 1.

$$E(C_t) = C_t + R * \pi_t \tag{1}$$

where $C_t$ is the production cost for unit $t$ in Table 1, $R$ is the probability of token confiscation, and $\pi_t$ is the predicted profit of the trade with no confiscation.

When the seller is the only party at risk of loss during conversion, the supply schedule becomes flatter, but results in the same predicted equilibrium as the No Seizure treatment (price of 80 tokens and a quantity tunnel between 20 and 24 tokens; Table 2; Fig 1). The expected profit for a buyer is unchanged from the No Seizure treatment at 150 tokens. The change in the effective supply schedule leads to a decrease in expected earnings for sellers compared to the No Seizure treatment. A seller is expected to earn 120 tokens in this treatment, compared to 150 in the No Seizure treatment. Due to this expected decrease in seller profit, the market surplus is expected to be 1,080, a slight decrease from the 1,200-token surplus expected in the No Seizure treatment.

When the buyer is the only party at risk of loss when converting tokens to dollars (Buyer Profit Seizure treatment), the demand schedule, and not the supply schedule, is affected. The new effective demand schedule is described by Eq 2.

$$E(RV_t) = RV_t - R * \pi_t \tag{2}$$

Where $RV_t$ is the redemption value of unit $t$. The risk to buyers leads to the same predicted equilibrium price and quantity as that in the Seller Profit Seizure and No Seizure treatments,

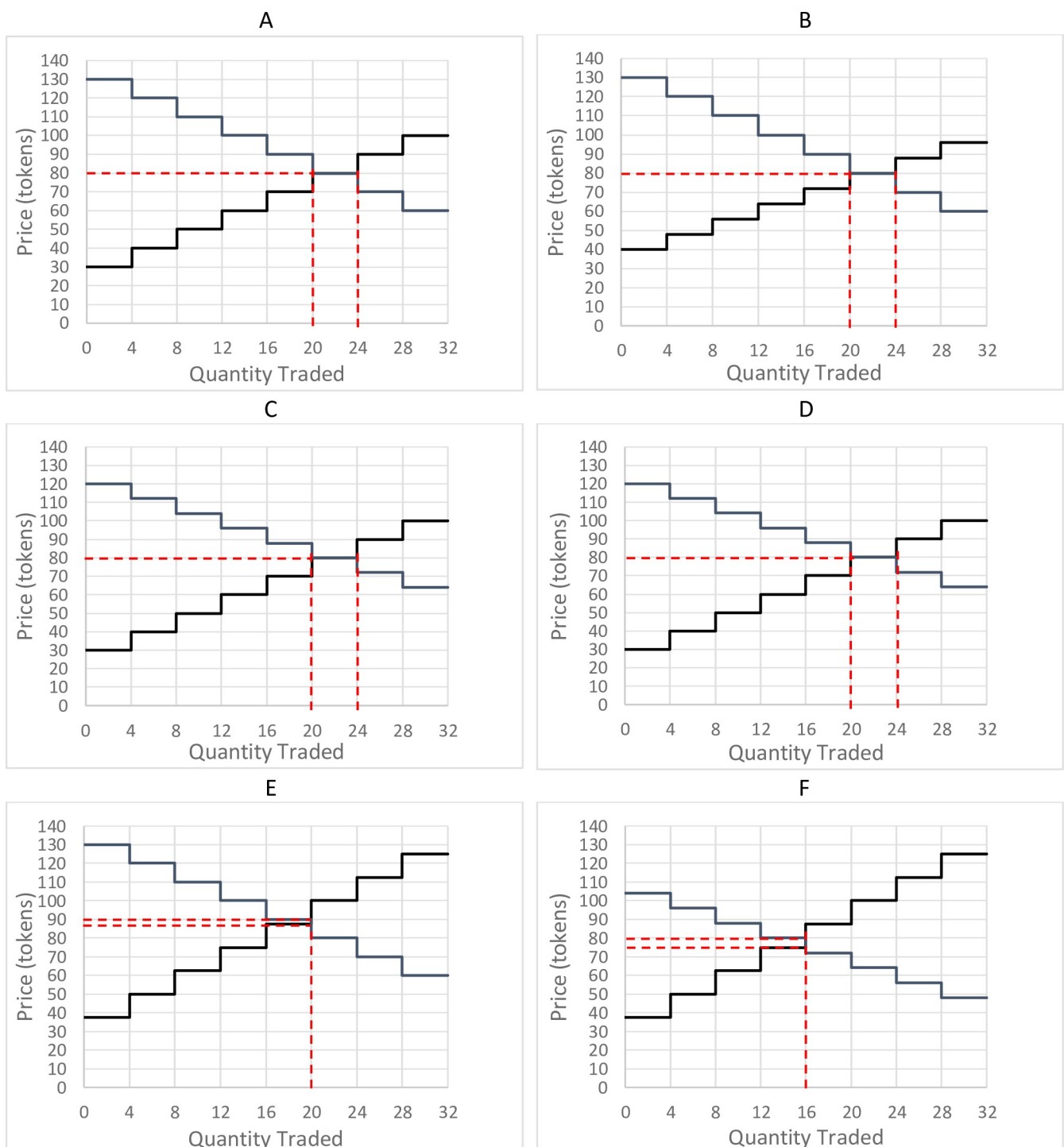

**Fig 1. Predicted market equilibrium across treatments.** (A) Predicted equilibrium in the No Seizure treatment. (B) Predicted equilibrium in the Seller Profit Seizure treatment. (C) Predicted equilibrium in the Buyer Profit Seizure treatment. (D) Predicted equilibrium in the Both Profit Seizure treatment. (E) Predicted equilibrium in the Product Seizure treatment. (F) Predicted equilibrium in the Trade Seizure treatment.

Table 2. Predicted market outcomes across treatments.

| Variable | Treatment | | | | | |
|---|---|---|---|---|---|---|
| | No Seizure | Seller Profit Seizure | Buyer Profit Seizure | Both Profit Seizure | Product Seizure | Trade Seizure |
| Units Produced /Traded | 20–24 | 20–24 | 20–24 | 20–24 | 20 | 20 |
| Price | 80 | 80 | 80 | 80 | 88–90 | 87.5–88 |
| Seller Earnings | 150 | 120 | 150 | 120 | 127.5–137.5 | 125–127.5 |
| Buyer Earnings | 150 | 150 | 120 | 120 | 100–110 | 80–82.5 |
| Total Earnings | 1200 | 1080 | 1080 | 960 | 950 | 830 |

and the same 1,080-token market surplus as in the Seller Profit Seizure treatment (Table 2; Fig 1). Yet, the shift in the effective demand schedule leads to a lower expected buyer profit of 120 tokens, while seller profit remains at 150, same as in the No Seizure treatment.

When both buyers and sellers are at risk of losing tokens when converting to a dollar currency (Both Profit Seizure treatment), the effective supply and demand schedules are determined by Eqs 1 and 2, respectively. The predicted equilibrium remains unchanged from the No Seizure treatment and other profit seizure treatments at a price of 80 and a quantity tunnel between 20 and 24 units (Table 2; Fig 1). Yet, both buyers and sellers see a reduction in expected profits compared to the No Seizure treatment. Both buyers and sellers are expected to have an individual profit of 120 tokens, generating a total market surplus of 960 tokens.

A risk of production loss prior to trade in the Product Seizure treatment, causes a change in the effective supply schedule. Since the loss happens before a trade is initiated with a buyer, the demand schedule remains unchanged. Given a risk of loss on every unit produced by 20%, the expected cost of producing a unit for trade increases. Following Lamb et al. [45], the effective supply schedule can be represented by:

$$E(C_t) = C_t * \eta_t \tag{3}$$

where

$$\eta_t = \frac{1}{1 - R} \tag{4}$$

and where $R$ is the probability of unit loss.

As a result of the shift in the effective supply schedule, the new equilibrium is a quantity of 20 units and a price tunnel between 88 and 90 tokens (Table 2; Fig 1). Compared to the No Seizure treatment, total surplus available in this treatment decreases to 950 tokens. Due to the increased slope of the supply schedule, the predicted earning of sellers is greater than that of buyers. The expected earnings for a buyer decrease to between 100 and 110 tokens, while the expected earnings of a seller ranges between 127.5 and 137.5 tokens.

In the case where a unit is lost after a trade is successfully negotiated (Trade Seizure treatment), both buyers and sellers are affected. The effective supply curve is again the same as in the Product Seizure treatment (Eqs 3 and 4) since the risk of a unit failing is constant across the two treatments. However, unlike production seizure, demand is affected by the risk of losing a unit after a trade is negotiated. In this treatment, if a trade is negotiated, but the unit is confiscated, then the buyer loses the potential profit of that trade. With a risk of a unit being lost after a trade is negotiated, the effective demand schedule decreases, as represented by Eq 3:

$$E(RV_t) = \pi_t(1 - R) + P^* \tag{5}$$

where $P^*$ is the predicted equilibrium price. Based on Eqs 3–5, and a 20% risk of loss, the effective supply and demand schedules for the Trade Seizure treatment generate the new predicted

equilibrium quantity of 20 units, while the price tunnel ranges between 87.5 and 88 tokens (Table 2; Fig 1). Due to the shift in both the supply and demand schedules, the total surplus available for this market decreases to 830 tokens. Since the effective demand schedule is flatter than the effective supply schedule, the expected profit for a seller of between 125 and 127.5 tokens is more than the expected profit for a buyer (80 to 82.5 tokens).

These simple comparative static exercises suggest that public policy involving the seizure of financial assets at the money laundering stage of the production-trade-laundering cycle should have no impact on the quantity of illegal products traded. Seizure of financial assets at this point is nothing more than a tax on profits, and as such there is no expected impact on the market equilibrium. This observation suggests if the intended goal is to curb illegal market activity, resources should be directed elsewhere in the production-trade-laundering cycle since the analytics suggest the more effective method is capturing the product at the point of trade or when it is produced.

While these simple comparative statics are instructive, previous research shows that agent behavior often results in market outcomes that can deviate substantially from theoretical predictions, especially when there is a risk of loss [45]. Thus, we conduct and analyze data generated from our laboratory market experiments to further illuminate the impact and efficacy of law enforcement actions on illicit trade.

## Analysis

Data on price, quantity of units produced, quantity of units traded (after seizure), seller earnings, buyer earnings, and total earnings (all earnings reported are after any profit seizure) for each trading period and treatment are collected. We analyze data using a parametric convergence model developed by Noussair et al. [46] to estimate each variable's convergence level for each treatment, as is common in the literature (e.g. [29–32,41]). The model is as follows:

$$Z_{it} = B_0 \left[ \frac{(t-1)}{t} \right] + B_1 \left( \frac{1}{t} \right) + \sum_{j=1}^{i-1} \alpha_j D_j \left[ \frac{(t-1)}{t} \right] + \sum_{j=1}^{i-1} \beta_j D_j \left( \frac{1}{t} \right) + u_{it} \qquad (6)$$

where $Z_{it}$ is the variable of interest such as quantity traded after seizure or earnings after seizure across treatment, $i$, for each trading period, $t$ (up to 20); $B_0$ is the predicted asymptote and $B_1$ is the starting level for the baseline treatment (No Seizure); $\alpha_j$ and $\beta_j$ are adjustments to asymptotes and starting levels in each $j$th treatment in relation to the baseline; and $D$ is the dummy variable representing the $j$th treatment (equal to zero for the No Seizure treatment and one for all other treatments).

The convergence model provides a statistical description of the path of the data from the beginning level to the asymptote of each treatment while addressing econometric issues found in the experimental data (such as heteroskedasticity, contemporaneous correlation, and serial correlation) [46]. As $t$ increases, the weight of the starting value becomes smaller, while the weight of the asymptote becomes larger. Through this analysis, we can evaluate treatment effects by testing for differences in the estimated asymptotes for each treatment without making *a priori* judgements on the appropriate data that represents stable convergence, such as is needed in more simplistic difference of means tests or other tests that do not account for $t$.

Panel data are typically both serially and contemporaneously correlated, and residuals for this type of dataset may not be homoscedastic [47]. We employ the Parks estimation technique to account for these issues simultaneously [47]. Given we use a random stop, individual experimental sessions range between 20 and 24 periods. In order to have a balanced panel, which is required by the Parks method, we truncate the data at 20 periods. By averaging across the

sessions in each treatment we avoid the potential contemporaneous correlation across sessions within treatments.

We evaluate statistically significant differences between estimated values from the convergence analysis across treatments to address our research objective. We use two-tailed t-tests to test for differences between asymptotic parameter estimates (estimated values the data converge to in period 20) across treatments. Since t-tests require data to be normally distributed [48], we conduct Shapiro-Wilk tests to determine normality at the 5% significance level for each variable. If a variable is not normally distributed and severely skewed as defined by Brown [49], the Wilcoxon's non-parametric Rank-Sum statistic is used (see [48]). All convergence analyses are conducted using SAS statistical software.

To lend validity to the convergence analysis, results are compared to two-tailed Wilcoxon's non-parametric Rank-Sum tests and regression analyses over the last five periods (which is used in the previous literature to represent convergence [30,31,50]) when including covariates found to potentially influence market outcomes from previous literature (S2 Text). Although including covariates when measuring treatment effect leads to biased estimators [51], recent literature suggests that properly adjusting for any imbalanced prognostic variables is appropriate [52,53]. These additional analyses are performed via STATA statistical software.

## Results and discussion

Prior to the start of the experiment, participants took part in at least one practice period. The average number of practice periods varied slightly between the treatments (Table 3), but not enough to expect a behavioral effect on treatment differences. Participants' gender was also collected since previous literature shows that it can affect individual outcomes in similar market experiments [41]. Every attempt was made to fully randomize participants across the treatments. Yet, based on the method outlined by Imbens and Rubin [51], our sample is imbalanced in respect to gender. As a result, gender (defined as the proportion of women participants in a session) is included as a covariate in regression analyses that are compared to the convergence analysis results (Table B in S2 Text).

Treatment effects for Units Produced provide an indication of how law enforcement focused on different stages along the production-trade-laundering cycle affects the amount of illegal product on the market, while the treatment effects for Units Traded provide an estimate for the amount of illegal trade. Other variables, such as Price, Buyer Earnings, Seller Earnings, and Total Earnings, also provide information on illegal product market outcomes (see S3 Text for data used in the analyses). Using these main variables, we next discuss the main findings to help address our research aim.

### Main result 1: Seizure during laundering of illicit profits does little to reduce volume of illegal production or trade

Based on the convergence model results, the quantities of units produced when profits are seized are not different when compared to no enforcement (Table 4). There is no difference in

**Table 3. Treatment-level statistics on number of practice periods and gender.**

| Variable | Treatment | | | | | |
| --- | --- | --- | --- | --- | --- | --- |
| | No Seizure | Seller Profit Seizure | Buyer Profit Seizure | Both Profit Seizure | Product Seizure | Trade Seizure |
| **Average Number of Practice Periods** | 1.5 | 1.33 | 1.17 | 1.5 | 1.67 | 1.67 |
| Min | 1 | 1 | 1 | 1 | 1 | 1 |
| Max | 3 | 2 | 2 | 2 | 2 | 2 |
| **Proportion of Women Participants** | 0.5 | 0.43 | 0.33 | 0.35 | 0.41 | 0.52 |

**Table 4. Estimates for market outcomes across treatments.**

| Variable | Treatment | | | | | |
|---|---|---|---|---|---|---|
| | **No Seizure** | **Seller Profit Seizure** | **Buyer Profit Seizure** | **Both Profit Seizure** | **Product Seizure** | **Trade Seizure** |
| **Units Produced** | 15.33 | 15.17 | 14.20 | 16.27 | 15.90 | 12.03* |
| **Units Traded** | 14.66 | 15.03 | 13.61* | 15.90* | 12.51* | 10.01* |
| **Price** | 75.19 | 73.18 | 72.15* | 77.17* | 79.67 * | 73.31 |
| **Seller Earnings** | 99.15 | 84.82* | 92.45 | 90.53 | 62.60* | 45.95* |
| **Buyer Earnings** | 143.21 | 141.91 | 115.91* | 109.96* | 112.14* | 101.29* |
| **Total Earnings** | 968.28 | 908.52* | 833.88* | 802.08* | 698.57* | 587.56* |

* Indicates a significant difference between the given treatment and the No Seizure treatment at the 5% significance level.

[i] All treatments have six replications.

[ii] Units Produced do not meet normality and are severely skewed as per Brown [49]. Thus, following previous research, we report averages for the last 5 trading periods per treatment and non-parametric test results [30,31,50]. All other variables meet the normality assumption and the convergence results and parametric tests are reported using data from all 20 trading periods.

units produced between the No Seizure treatment (15.33), the Seller Profit Seizure (15.17), the Buyer Profit Seizure (14.20), and Both Profit Seizure (16.27) treatments ($p$-value = 0.7338, 0.1970, and 0.3939, respectively). These results are consistent with Wilcoxon's non-parametric Rank-Sum tests (Table A in S2 Text) and the regression analysis (the only difference to note is that the regression analysis found that the Both Profit Seizure treatment had significantly more units produced than the No Seizure treatment; Table C in S2 Text).

We do see some statistically significant differences in units traded when profits are seized as compared against no law enforcement. However, the results overall do not suggest major reductions in illicit trade. The number of units traded is slightly lower in the Buyer Profit Seizure treatment, but the less than one-unit difference is statistically significant (13.61; $p$-value < 0.0001). There is no statistical difference in units traded when comparing Seller Profit Seizure (15.03; $p$-value = 0.3034) to the No Seizure treatment (14.66). The number of trades is actually higher in the Both Profit Seizure (15.90; $p$-value = 0.0056), suggesting that money laundering enforcement aimed at both the buyer and seller of illegal products, may actually *increase* the amount of illegal trading compared to no enforcement at all. Again, these results are consistent with the regression analysis and Wilcoxon's non-parametric Rank-Sum tests (Table A and C in S2 Text).

Seizure of assets at the money laundering stage of the production cycle, for the probability levels modeled here, have little to no impact on reducing the production and distribution of illegal products. Only when buyers' money laundering attempts are at risk of law enforcement, is distribution expected to slightly decrease. If the goal of seizure of illicit profit is to reduce the amount of illegal production and market activity from which the profit was created, then our results show this form of law enforcement is relatively ineffective.

## Main result 2: Seizure during trade is most effective at reducing illegal products on the market

The amount of illegal product on the market after law enforcement action is lowest in the Trade Seizure treatment (Table 4). This risk of product seizure to both the buyers and sellers in this treatment reduces the amount of product on the market. Compared to the No Seizure treatment, the number of units traded is nearly 32% lower in the Trade Seizure treatment, 14.66 versus 10.01 units, respectively ($p$-value < 0.0001). The only other treatments in which the amount traded is lower than no enforcement is when enforcement is aimed at confiscation during production or when buyers are at risk of profit seizure. Units successfully traded is 15%

lower in the Product Seizure treatment (12.51 units; $p$-value < 0.0001) and 7% lower in the Buyer Profit Seizure treatment (13.61 units; p-value < 0.0001) than the No Seizure treatment.

The reduced quantity of product traded in the Trade Seizure treatment compared to the No Seizure treatment (10.01 versus 14.66, respectively; $p$-value < 0.0001) is consistent with the greatest reduction in the amount of product initially produced. Nearly 22% fewer units are produced under the Trade Seizure treatment than in the No Seizure treatment (12.03 versus 15.33 units, respectively; $p$-value = 0.0433). The confiscation of units at the time of trade results in loss of production costs for the seller, which reduces supply. The simultaneous decrease in demand creates an incentive to produce fewer units overall.

The cause of reduced trade in the Product Seizure treatment compared to the No Seizure Treatment (12.51 versus 14.66, respectively) is apparently not related to a reduction in the amount produced: the quantity of units produced in the Product Seizure treatment (15.90) is not statistically different from the No Seizure treatment ($p$-value = 0.8182). In the Product Seizure treatment, the higher prices and potential for profit from traded units mitigate risk of loss from product seizure, and encourage production levels similar to those in the No Seizure treatment. Thus, the incentives generated from law enforcement focused on seizure at the time of trade (but not during production) will likely not only reduce the amount of illegal product traded, but also reduce the amount produced. In the immediate short run, law enforcement resources focused on seizure during trading is expected to lead to the most reduction in illegal market activity. These results are supported by the regression and Wilcoxon's non-parametric Rank-Sum tests, lending further support to these outcomes (Table A and C in S2 Text).

The Trade Seizure treatment also gives implications for the long run. Seller, buyer, and total market earnings are the lowest in the Trade Seizure treatment (Table 4). Since both price and quantity traded are at the lowest level in this treatment, seller earnings are nearly 50% less than when there is no law enforcement (45.95 versus 99.15, respectively; $p$-value < 0.0001, consistent with regression and Wilcoxon's non-parametric Rank-Sum tests; Table A and C in S2 Text). Although buyers have drastically higher profits in this treatment compared to sellers (120% more), buyers are still the worst off when enforcement is focused on seizure during trade. As a result, total earnings are the lowest in this treatment, and are nearly 40% less than with no law enforcement ($p$-value < 0.0001, consistent with regression and Wilcoxon's non-parametric Rank-Sum tests; Table A and C in S2 Text). Since earnings are reduced by such a large margin, 25% of sellers in this treatment earned negative profits in these treatment sessions (i.e., their total payoff was less than the initial balance they were given to start the session). In the long run, these sellers would be expected to leave the market. When law enforcement seizes products during trade, there may be a spillover effect with sellers potentially leaving the market in the long run, which may further reduce illegal market activity and crime. This result is similar to other experimental studies that show effective enforcement focused on other sources of crime (e.g. tax evasion) have spillover effects that may further help in reducing crime [54–56].

These experimental results are somewhat consistent with those predicted from the comparative static results reported in Table 2; Trade Seizure was predicted to have the most impact on market outcomes overall. However, the observed prices, quantities and earnings differ from predictions. Behavior changes the most, and the greatest reduction in market activity comes from seizure of goods at the point the goods are traded.

## Main result 3: Risk influences behavior leading to outcomes that vary from those predicted

Consistent with previous market experiments, bargaining outcomes differ from those predicted [29–32,40,45]. Comparing Tables 2 and 4, in all cases, prices, quantities, and total

**Table 5. Enforcement incidence by treatment (in percent).**

| Variable | Treatment | | | | |
|---|---|---|---|---|---|
| | Seller Profit Seizure | Buyer Profit Seizure | Both Profit Seizure | Product Seizure | Trade Seizure |
| **Predicted Seller Enforcement Incidence [i]** | 100 | 0 | 50 | 28 | 26 |
| **Predicted Buyer Enforcement Incidence [ii]** | 0 | 100 | 50 | 72 | 74 |
| **Realized Seller Enforcement Incidence [iii]** | 92 | 20 | 21 | 54 | 56 |
| **Realized Buyer Enforcement Incidence [iv]** | 8 | 80 | 79 | 46 | 44 |
| **Difference between Expected and Realized Incidence [v]** | 8 | -20 | 29 | -26 | -30 |

[i] Calculated as the expected loss in Seller Earning, divided by the expected loss in Total Earning (Seller and Buyer Earnings) in the given treatment compared to the No Seizure treatment. Values are based on the theoretical predictions presented in Table 2.

[ii] Calculated as the expected loss in Buyer Earning, divided by the expected loss in Total Earning (Seller and Buyer Earnings) in the given treatment compared to the No Seizure treatment. Values are based on the theoretical predictions presented in Table 2.

[iii] Calculated as the realized loss in Seller Earning, divided by the realized loss in Total Earning (Seller and Buyer Earnings) in the given treatment compared to the No Seizure treatment. Values are based on the convergence analysis presented in Table 4.

[iv] Calculated as the realized loss in Buyer Earning, divided by the realized loss in Total Earning (Seller and Buyer Earnings) in the given treatment compared to the No Seizure treatment. Values are based on the convergence analysis presented in Table 4.

[v] Calculated as the difference between expected and realized incidence for sellers for the given treatment. A positive value indicates a reduction in realized incidence, compared to theoretical predictions, for sellers (a benefit to sellers). A negative value indicates a reduction in realized incidence, compared to theoretical predictions, for buyers (a benefit to buyers).

earnings are below the levels predicted. Importantly, similar to our results, previous research has shown reduced bargaining power of sellers when trades are privately negotiated, resulting in fewer units produced, lower prices, lower seller earnings, and higher buyer earnings than predicted by theory [29,30,41]. The increased bargaining power of buyers can be further seen by the amount of the cost of enforcement absorbed by sellers compared to buyers. Based on the theoretical predications presented in Fig 1 and Table 2, the incidence of enforcement (the percent decrease in Total Earnings from enforcement absorbed by the relevant party) for both sellers and buyers are shown in Table 5. Based on theory, sellers should only have a higher enforcement incidence than buyers when enforcement is aimed at money laundering of sellers only (Table 5). Yet, the experimental results show that sellers additionally absorb more enforcement cost than buyers in the Product Seizure and Trade Seizure treatments (the percent decrease in Total Earnings absorbed by the relevant group from Table 4 and shown in Table 5). Further, comparing the predicted seller enforcement incidence to the realized (from the experimental data) seller enforcement incidence in Table 5, sellers have a higher enforcement incidence than expected in the Buyer Seizure, Product Seizure, and Trade Seizure treatments (Row 5). Thus, buyers are able to use their increased bargaining power in these treatments to push some of their cost of enforcement onto sellers.

It is only in the Seller Profit Seizure and Both Profit Seizure treatments that sellers are able to pass on some of their expected loss in earnings to buyers (Table 5, Row 5). But in neither of these treatments were sellers able to pass on enough of the cost of enforcement for sellers to have higher earnings than buyers (Table 4). As such, buyers seem to have a bargaining advantage over sellers in illegal product markets.

Table 5 also shows that theoretical predications of incidence, which depend solely on the relative elasticity of supply and demand [57], may fail to hold for enforcement incidence in bilateral bargaining markets. This result is similar to other experimental studies of incidence (e.g. [33]), especially in markets with bilateral bargaining [31,32], since trading institution has been shown to impact incidence [58].

Further, the quantity of units produced and traded are well below those predicted in all treatments (Tables 2 and 4). Even with no probability of seizure, the amount produced is 23 to 36% less, and the amount traded is 20 to 39% less than predicted. The amount produced deviates furthest from predictions in the Trade Seizure treatment, with 40% less produced and 50% less traded than expected. The risk of seizure at the point of trade induces sellers to substantially reduce the amount they produce and trade, compared to equilibrium predictions.

In all treatments, buyer earnings are higher than seller earnings, reinforcing the observation that the bargaining power of buyers is greater than sellers in this market (buyers earn 9 to 125% more than sellers depending on the treatment; Table 4). Although buyer earnings are typically below their predicted values, they are much closer to predicted values than seller earnings. Seller earnings are always lower than predicted. In the Product Seizure treatment, seller earnings are less than half of what was predicted. In the Product Seizure treatment, buyer earnings are actually higher than what was predicted. Buyer earnings do not exhibit a similar pattern as seller earnings across treatments. In the Product and Trade Seizure treatments, buyers do not risk loss of an already sunk investment. As a result, there is not systematic difference across treatments.

Due to the fact that both buyer and seller earnings are typically below that predicted, total earnings are far below expected outcomes. Even with no seizure presence, total earnings are found to be 19% below that predicted. Among the seizure treatments, the Trade Seizure treatment sees the largest difference between experimental results and predicted outcomes, with nearly 30% less total earnings than expected (Table 4).

There is also a stark reduction in seller and total earnings in the Trade and Product seizure treatments relative to the Profit Seizure treatments, likely due to the different mechanism of loss. In the Trade and Product Seizure treatment, sellers risk losing all incurred production costs of units seized. In the Seller and Both Profit seizure treatments, sellers only risk losing a portion (20%) of earnings. The differences in risk are likely reducing production and trade levels in the Trade Seizure treatment, reducing trades in the Production Seizure treatment, and thus exacerbates the reduction in seller earnings compared to the Profit Seizure treatments.

Since experiment outcomes differ from predictions, law enforcement policy built upon theoretical findings alone may lead to unexpected outcomes and effectiveness when attempting to combat illicit trade.

## Conclusions

Despite national and international policies aimed at reducing illicit trade, illegal product markets account for two to five percent of global GDP. Although many efforts have focused on anti-money laundering, law enforcement aimed at reducing illicit trade can also take place at (1) the production of the illegal product, and (2) sale of the illegal product. To provide guidance on where in this production-trade-laundering cycle law enforcement is most impactful at reducing illicit trade, we use an experimental market environment that mimics these three enforcement foci.

Our results suggest that focusing law enforcement on seizing laundered profits is less of a deterrent than other enforcement policies analyzed. Seizing profit from either only the seller or from both buyer and seller, does not reduce the amount produced or sold when compared to no law enforcement present. When buyers are the only ones at risk of profit seizure, trade is found to decrease at a statistically significant level compared to no law enforcement, but not nearly as much as when enforcement is focused on seizure during the sale of illegal products. When profit was at risk of seizure, earnings did decrease compared to no law enforcement, but were higher than any other treatment with law enforcement. These results suggest that

focusing law enforcement on seizing illicit profits through money laundering will likely not create incentives to reduce illicit trade.

Our results also suggest that illicit trade is likely reduced the most when law enforcement is aimed at product seizure at the point of sale, or during a trade. Using a seizure rate of 20%, the amount of illicit trade was found to be nearly 32% lower when law enforcement is focused at the point of sale when compared to no law enforcement, with no risk of seizure. This risk of seizure during trade is found to incentivize participants to reduce illicit trade by more than the amount of that seized by law enforcement. Further, since both buyer and seller earnings are found to be the lowest under this form of law enforcement, and a quarter of sellers incurred profit losses, there may be additional economic incentives for buyers and sellers to leave the market, potentially causing further reduction in illicit trade in the long run.

Similar to previous literature, market outcomes in this experimental market differ from theoretical predictions. When the market institution is bilateral negotiations and sellers must have inventory of illegal product on hand that cannot be carried over from season to season, prices and trades are lower than predicted. Buyers are in a relatively better bargaining position than sellers, and buyers tend to earn more than sellers in these market environments. Policies against illegal sales not only impact the total amount traded, but also affect the relative earnings of sellers and buyers. Trade Seizure has the greatest impact on total market activity and also impacts seller earnings more than buyer earnings. This impact increases the opportunity cost of staying in the market and gives the greatest probability of sellers exiting. These empirical deviations from equilibrium predictions suggest creating law enforcement policy based on theory alone may lead to inefficient allocation of limited law enforcement resources.

Research regarding optimal law enforcement needs to understand deterrence of criminal activity across different possible punishment strategies given budget constraints. Yet, no study compares the effectiveness of multiple law enforcement policies aimed at illegal market activity and money laundering while accounting for individual behavior. The current literature offers few insights regarding how policies may interact with observed market behavior, and thus, may cause enforcement mechanisms to deviate from theory. We address this research gap. At the time of this writing, credible data indicating actual seizure rates relative to a total market were not available.

It is unlikely that law enforcement agencies would be able to actually seize 20% of the product or profits of the illegal products. The seizure level modeled here provides guidance on where to concentrate law enforcement resources in order to best reduce the size of the market. Further, the buyers and sellers modeled in this study are assumed to represent large entities where law enforcement, even when successful, cannot seize all illegal assets at any stage along the production-trade-laundering cycle. It is thus assumed that operations modeled are large enough, for instance, that if one field of illegal crop is eradicated, there are many other fields left undetected, or illicit profit is laundered through multiple avenues, such that, even with detection of money laundering in some avenues, other avenues remain undetected. Yet, the potential exists for law enforcement to successfully seize all assets of smaller operations than assumed here. Additionally, our analysis does not capture the ability of operations to reinvest profits into capital acquisition to expand production capabilities or invest in other criminal activities. Future research should focus on how effective various law enforcement policies are in such situations.

Our results suggest that seizure of product at the point of sale creates the most impactful deterrence of illegal product market activity, and these impacts are different than those predicted for a competitive market. Additionally, our results suggest that seizure of laundered profits are least impactful in terms of deterring production and trade in an illegal market.

These results suggest that all else equal, a dollar spent on trade seizure is likely to be more of a deterrence than a dollar spent on seizing during production or laundering of funds.

## Supporting information

**S1 Text. Experimental instructions.**
(DOCX)

**S2 Text. Wilcoxon's non-parametric Rank-Sum and regression results.**
(DOCX)

**S3 Text. Data and SAS and STATA code.**
(DOCX)

## Acknowledgments

We would like to thank Shaya Wolf for her programming of the computer program used in the laboratory market experiment for this study.

## Author Contributions

**Conceptualization:** Chian Jones Ritten, Christopher Bastian, Owen Phillips.

**Data curation:** Chian Jones Ritten, Christopher Bastian, Owen Phillips.

**Formal analysis:** Chian Jones Ritten, Christopher Bastian.

**Funding acquisition:** Christopher Bastian, Owen Phillips.

**Methodology:** Chian Jones Ritten, Christopher Bastian, Owen Phillips.

**Project administration:** Chian Jones Ritten.

**Writing – original draft:** Chian Jones Ritten, Christopher Bastian, Owen Phillips.

**Writing – review & editing:** Chian Jones Ritten, Christopher Bastian.

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
