## [Decision Letter · Decision Letter 0]

16 Apr 2021

PONE-D-21-07819

Money Laundering, Crime, and the Seizure of Assets: Relative Effectiveness of Law Enforcement Policies in Experimental Bargaining Markets

PLOS ONE

Dear Dr. jones ritten,

Thank you for submitting your manuscript to PLOS ONE. After careful consideration, we feel that it has merit but does not fully meet PLOS ONE’s publication criteria as it currently stands. Therefore, we invite you to submit a revised version of the manuscript that addresses the points raised during the review process.

Both reviewers found the study well written and recommend the paper to be revised. The reviewers provide constructive criticism which they ask you to carefully consider, and address in your revision. I have read the paper myself and I can only agree with their evaluation. In addition to their criticism, I would like to raise a few further points I would like to you to pay attention to in your revision

1) You mention in the paper that practice periods were run up to the point where no participant asked for further practice rounds. I would like you to very briefly report some treatment level statistics on the number of practice rounds as this could have an effect on treatment differences. Such variation is not a reason to reject the paper, but it is valuable information for the reader when evaluating the mechanisms behind the treatment effects.

2) The trading mechanism is repeat bilateral bargaining with an exogenously set number of matches (3). This procedure differs from the more standard practices of double auction markets. I would like you to discuss briefly, referring to existing literature if necessary, why you think the competitive equilibrium (which you use in deriving your predictions) should provide accurate predictions for the this market mechanism.

3) The competitive equilibrium prices and quantities depend on the slopes and and intercepts of the demand and supply schedules. I would like to better understand, how your treatments are hypothesized to effect on these slopes and intercepts, e.g. on the expected marginal costs of the seller. A figure depicting these schedules in each treatment might be useful, maybe condensed into one figure or a figure with several panes to allow for easy comparison of the predictions.

We look forward to receiving your revised manuscript.

Kind regards,

Topi Miettinen, PhD

Academic Editor

PLOS ONE

Journal Requirements:

Please consider changing the title so as to meet our title format requirement (https://journals.plos.org/plosone/s/submission-guidelines). In particular, the title should be "Specific, descriptive, concise, and comprehensible to readers outside the field" and in this case it is not informative and specific about your study's scope (in particular the 'lab' setting) and methodology.

3, Please include captions for *all* your Supporting Information files at the end of your manuscript, and update any in-text citations to match accordingly. Please see our Supporting Information guidelines for more information: http://journals.plos.org/plosone/s/supporting-information.

Reviewers' comments:

Reviewer's Responses to Questions

**Comments to the Author**

1. Is the manuscript technically sound, and do the data support the conclusions?

Reviewer #1: Yes

Reviewer #2: Partly

2. Has the statistical analysis been performed appropriately and rigorously? 

Reviewer #1: Yes

Reviewer #2: I Don't Know

3. Have the authors made all data underlying the findings in their manuscript fully available?

Reviewer #1: Yes

Reviewer #2: No

4. Is the manuscript presented in an intelligible fashion and written in standard English?

Reviewer #1: Yes

Reviewer #2: Yes

5. Review Comments to the Author

Reviewer #1: This paper aims to compare the effectiveness in trade reduction of three enforcement policies: profit seizure of either buyers, sellers or both, product seizure after production and product seizure after a trade is negotiated.

To address this question, the authors derived equilibrium predictions and designed a market experiment to test their main comparative statics in which they manipulate the enforcement institution.

The authors find that profit seizure policies do not reduce the volume of quantities produced or traded compared to the volume in the absence of regulation. In contrast, product seizure after trade has the greater impact on both quantity produced, and quantity traded.

The authors contribute to the literature by enhancing our understanding of where in the production-trade-laundering cycle, enforcement policies are the most effective at curbing illegal trade. The experimental design enables clean comparisons between the three policies in terms of quantities produced and traded, prices and profits.

Nonetheless, the paper would benefit from some clarifications.

1. From the introduction, the reader is led to believe that most enforcement policies aiming at curbing illegal trade currently in place are targeting the money laundering stage of the cycle. However, there are often reports in the news of drugs seizure after trade were negotiated (e.g. during transport) or after production. From what I understand, the problem is that enforcement currently takes place anywhere along the production-trade-laundering cycle but there is no evidence yet on where along this cycle enforcement is the most efficient. If this is the case, I would advise the authors to reorganize their introduction along these lines and to not overextend on policies aiming at reducing money laundering only. If it appears that most of the effort is concentrated towards the laundering stage of the cycle, then it poses the question of why it is the case: is it because it is more difficult to implement policies at earlier stages of the cycle? A discussion on these aspects seems highly relevant to appreciate the policy implications of the paper.

2. I cannot understand how to read the letters in Table 3 that denote the significance of pairwise comparisons between treatments. Perhaps there is a more intuitive way to do this? Otherwise, perhaps the authors could clarify how these letters should be interpreted.

3. Regarding Table 3 again, the authors indicate significance at the 10% level. Given the strong significance of the results, I would recommend the authors to be more conservative and only indicate significance at the 5% level. Doing so will not affect their main results and reassure the reader.

Reviewer #2: This paper examines the effect of law enforcement on outcomes in "illegal" markets using an lab experiment. It concludes that a policy based on seizure of profits has little effect but targeting the point of sale is effective.

1) The characteristics of the participants in each treatment need to be presented and tests of equality/balance carried out.

2) Seizure as a policy in the "real world" entails costs over and above the amount seized. As law enforcement detects illegal activity and the channels through which the profits are laundered, criminals must hire new agents or create new ways to launder their money. There is also a chance that they will lose all profits and exit the game entirely. I think this needs to be acknowledged as seizure of profits in the experiment simply sees the "criminals" lose tokens with some probability.

3) Are the participants equally distributed across the treatments?

4) I am unfamiliar with the parametric convergence model and I think the rationale for using it should be discussed. In any event, for transparency and completeness I would like to see the analysis starting with a simpler method of comparing means across treatments and running regressions with treatment effects (including any personal characteristics that vary by treatment).

6. PLOS authors have the option to publish the peer review history of their article (what does this mean?). If published, this will include your full peer review and any attached files.

Reviewer #1: No

Reviewer #2: No

---

## [Author Response · Author response to Decision Letter 0]

27 May 2021

Dear Dr. Miettinen,

Thank you for the opportunity to submit an improved revision of the manuscript originally titled, “Money Laundering, Crime, and the Seizure of Assets: Relative Effectiveness of Law Enforcement Policies in Experimental Bargaining Markets.” We sincerely appreciate the valuable comments provided by yourself and the two reviewers. We have attempted to address them all, and believe the manuscript is much improved because of them. Below is a detailed response to all the comments made by yourself and reviewers, specifying changes to the original manuscript to address the concerns. Changes to the manuscript are noted by blue in the text.

Dr. Miettinen:

I have read the paper myself and I can only agree with their evaluation. In addition to their criticism, I would like to raise a few further points I would like to you to pay attention to in your revision

1) You mention in the paper that practice periods were run up to the point where no participant asked for further practice rounds. I would like you to very briefly report some treatment level statistics on the number of practice rounds as this could have an effect on treatment differences. Such variation is not a reason to reject the paper, but it is valuable information for the reader when evaluating the mechanisms behind the treatment effects.

We agree that including treatment level statistics on the number of practices periods is important information for the reader. We have included a table including the summary statistics on the number of practice periods per session for each treatment in Table 3 on page 23. We further explain that due to the small variation in practice periods across treatments, we do not expect this to have an influence on treatment effects. We include the following to lines 497-499 on page 23 to explain this:

“Prior to the start of the experiment, participants took part in at least one practice period. The average number of practice periods varied slightly between the treatments (Table 3), but not enough to expect a behavioral effect on treatment differences.”

2) The trading mechanism is repeat bilateral bargaining with an exogenously set number of matches (3). This procedure differs from the more standard practices of double auction markets. I would like you to discuss briefly, referring to existing literature if necessary, why you think the competitive equilibrium (which you use in deriving your predictions) should provide accurate predictions for the this market mechanism.

Thank you for your comment. We use the competitive equilibrium only as a guide to understand potential policy impacts prior to testing. Our primary objective is not to test theoretical market outcomes, but to understand real-world market outcomes. Thus, we choose parallelism in our laboratory market design for illegal products and choose bilateral bargaining as the appropriate trading institution. 

Davis and Holt (1993) state that in markets where agents make decisions sequentially and in real time, trading institutions are much more difficult to analyze theoretically (including double auction and bilateral negotiation), but these institutions offer parallelism and are closer to institutional rules found in many real-world markets. The double auction often is used in experimental market research, as it is thought to yield results mimicking the competitive equilibrium. Yet, research investigating differences in trading institutions and delivery methods relevant in commodity markets finds that neither the double auction nor private negotiation yields precisely the competitive equilibrium (Menkhaus et al., 2003). Menkhaus et al. (2003) find that under the same supply and demand conditions as used in the current study, the double auction with forward delivery yields prices below and trades above the predicted competitive equilibrium. The double auction with spot delivery was found to have a converged price above the competitive equilibrium and trades equal to the equilibrium. Private negotiation with forward delivery and three bargaining matches yields prices above the competitive equilibrium and trades below the predicted equilibrium. Private negotiation with spot delivery yields price and trades below the competitive equilibrium. Thus, neither institution was found to generate the competitive equilibrium perfectly. 

As both Davis and Holt (1993) and Menkhaus et al. (2003) show, even though double auction is a common institution used in the literature, it also cannot be expected to yield the expected equilibrium in all cases. We do acknowledge that is has been found to produce market outcomes that are closer to the predicted equilibrium when compared to bilateral bargaining. To our knowledge, no illegal product market uses double auction and its formal trading rules. There is, therefore, a resulting trade-off between parallelism of illegal product markets and convergence to predicted outcomes with our choice of institution. For our purpose of measuring real-life market outcomes in illegal product markets, we feel that parallelism is more important. 

We do agree that additional language is necessary in the manuscript to further explain our reasoning for using bilateral bargaining instead of the double auction institution. We have added the following to lines 218-223 on page 10: 

“Many market experiments use a double auction institution, as it is thought to yield results mimicking the competitive equilibrium [36]. However, the market institution through which illegal products reach the consumer is bilateral bargaining. Given this reality, past research has used a bilateral bargaining market for illegal product sales (e.g. [37]). Since our intent is to understand market outcomes in real-world illegal product markets, we use bilateral bargaining, as it offers closer parallelism than the double auction to these markets [37].”

References:

Davis, D. D., and C. A. Holt. 1993. “Chapter 1: Introduction and Overview.” Experimental Economics. Princeton, New Jersey: Princeton University Press. Pp. 3-66.

Menkhaus, D. J., O. R. Phillips, and C. T. Bastian. 2003. “Impacts of Alternative Trading Institutions and Methods of Delivery on Laboratory Market Outcomes.” American Journal of Agricultural Economics. Vol 85. No. 5: 1323-1329.

3) The competitive equilibrium prices and quantities depend on the slopes and intercepts of the demand and supply schedules. I would like to better understand, how your treatments are hypothesized to effect on these slopes and intercepts, e.g. on the expected marginal costs of the seller. A figure depicting these schedules in each treatment might be useful, maybe condensed into one figure or a figure with several panes to allow for easy comparison of the predictions.

We agree that a figure including the schedules and predicted equilibria improves the paper. We have included Figure 1 on page 17 that depicts the effective supply and demand schedules along with the predicted equilibria by using panes for each treatment. 

We have changed file names that now meet the style requirements.

2) Please consider changing the title so as to meet our title format requirement. In particular, the title should be "Specific, descriptive, concise, and comprehensible to readers outside the field" and in this case it is not informative and specific about your study's scope (in particular the 'lab' setting) and methodology.

We have changed the title of the manuscript to:

“The Relative Effectiveness of Law Enforcement Policies Aimed at Reducing Illegal Trade: Evidence from Laboratory Markets”

3) Please include captions for *all* your Supporting Information files at the end of your manuscript, and update any in-text citations to match accordingly. 

We now include captions for each of the three Supporting Information files at the end of the edited manuscript.

Reviewer #1: 

This paper aims to compare the effectiveness in trade reduction of three enforcement policies: profit seizure of either buyers, sellers or both, product seizure after production and product seizure after a trade is negotiated. To address this question, the authors derived equilibrium predictions and designed a market experiment to test their main comparative statics in which they manipulate the enforcement institution.

The authors find that profit seizure policies do not reduce the volume of quantities produced or traded compared to the volume in the absence of regulation. In contrast, product seizure after trade has the greater impact on both quantity produced, and quantity traded.

The authors contribute to the literature by enhancing our understanding of where in the production-trade-laundering cycle, enforcement policies are the most effective at curbing illegal trade. The experimental design enables clean comparisons between the three policies in terms of quantities produced and traded, prices and profits.

Thank you

Nonetheless, the paper would benefit from some clarifications.

1. From the introduction, the reader is led to believe that most enforcement policies aiming at curbing illegal trade currently in place are targeting the money laundering stage of the cycle. However, there are often reports in the news of drugs seizure after trade were negotiated (e.g. during transport) or after production. From what I understand, the problem is that enforcement currently takes place anywhere along the production-trade-laundering cycle but there is no evidence yet on where along this cycle enforcement is the most efficient. If this is the case, I would advise the authors to reorganize their introduction along these lines and to not overextend on policies aiming at reducing money laundering only. If it appears that most of the effort is concentrated towards the laundering stage of the cycle, then it poses the question of why it is the case: is it because it is more difficult to implement policies at earlier stages of the cycle? A discussion on these aspects seems highly relevant to appreciate the policy implications of the paper.

Thank you for this very meaningful comment. Our intent was to show that although there are various law enforcement policies along the production-trade-laundering cycle, currently, there is a push for policies to focus on the money-laundering stage. We have added and edited text in the abstract and the introduction to better reflect our intention. Specifically, the abstract now includes the following:

“Despite recent emphasis and implementation of national and international anti-money laundering policies, illegal product markets, and their associated illicit profit remain a global problem. In addition to law enforcement aimed at reducing money-laundering, enforcement also takes place during (1) the production (e.g. crop eradication) and (2) sale (e.g. seizure of products during transportation that interrupts buyer and seller transactions) of the illegal product.” 

The introduction has been expanded and now includes the following to lines 58-81 on pages 3-4: 

“Globalization has provided new opportunities for the proliferation of illegal product distribution. As a result, many international policies exist to halt the production and distribution of illegal products, and the laundering of illicit profit. These policies mainly use the seizure of assets throughout the production-trade-laundering cycle of illegal product markets to attack the financial structure and money laundering of groups worldwide [4]. Other penalties, including fines and prison terms, can be conceptualized as a seizure of assets from a broader perspective. Asset seizure and forfeiture is popular because it can reduce trade activity and profits (and therefore punish) of all entities involved, even though it may be difficult to identify all involved parties [5]. In 2020, nearly $600 million worth of illegal drugs and related assets were seized in the United States alone [4]. 

Various policies focus on seizing the production of illegal products before they are sold to consumers. For instance, the Drug Enforcement Administration (DEA) uses crop eradication to reduce the quantity of illegal drugs before they can hit the market. In 2019, the DEA eradicated over 4 million marijuana plants and seized nearly $30 million dollars of cultivator assets [6].

In addition to policies that eradicate illegal products to reduce the amount of product on the market, policies also aim to capture assets during the sale of these illegal products. These policies include street-level enforcement to interrupt transactions and border enforcement [7]. Border enforcement focuses on seizing illegal products as they cross international boundaries. Since most of these products are in transit to their buyer, seizure at the border, like street-level enforcement, aims law enforcement actions during the sale or trade of illegal products. 

Although law enforcement polices exist that focus on the production or sale of illegal products, recent emphasis has been placed on heightened enforcement aimed at the laundering of illicit profit.”

We have also added the following to lines 90-91on page 4:

“As a result of the recent spotlight on money laundering, numerous national and international anti-money laundering policies and entities have been created over the last 40 years.”

We have also added the following to lines 117-120 on page 6:

“Since funds for enforcement come from limited budgets, focusing enforcement in one stage of this cycle, likely results in less enforcement available for the other stages [15]. Thus, understanding where in this production-trade-laundering cycle law enforcement is most impactful becomes a global question.”

2. I cannot understand how to read the letters in Table 3 that denote the significance of pairwise comparisons between treatments. Perhaps there is a more intuitive way to do this? Otherwise, perhaps the authors could clarify how these letters should be interpreted.

Thank you for highlighting the confusion of the letters used in the table. We have edited the table to now only include statistical differences of the treatments to the base (No Seizure treatment). We now only include * to indicate a significant treatment effect with respect to the No Seizure treatment. We think this change addresses the confusion you indicate and does not lose any vital information.

3. Regarding Table 3 again, the authors indicate significance at the 10% level. Given the strong significance of the results, I would recommend the authors to be more conservative and only indicate significance at the 5% level. Doing so will not affect their main results and reassure the reader.

This is a great point, and we have updated Table 4 to only indicate significant treatment effects at the 5% level.

Reviewer #2: 

This paper examines the effect of law enforcement on outcomes in "illegal" markets using an lab experiment. It concludes that a policy based on seizure of profits has little effect but targeting the point of sale is effective.

1) The characteristics of the participants in each treatment need to be presented and tests of equality/balance carried out.

Every attempt was made to randomize participants across the treatment. We do recognize that with experiments such as ours, our attempts may still not lead to full randomization of participants over the treatments. Based on a previous study showing that gender may impact individual outcomes in these types of laboratory market experiments, we collected the gender of participants. We have now added the proportion of women per session across the treatments in Table 3 on page 23. 

We have also included a test of balance of participants’ gender across treatments using the method outlined by Imbens and Rubin (2015), which is included in S2 Text. We do find that gender is imbalanced across the treatments. We have added statistics on the proportion of women per session in Table 3 on page 23, and added the following text in the main manuscript to lines 499-504 on page 23 to state our finding:

“Participants’ gender was also collected since previous literature shows that it can affect individual outcomes in similar market experiments [41]. Every attempt was made to fully randomize participants across the treatments. Yet, based on the method outlined by Imbens and Rubin [51], our sample is imbalanced in respect to gender. As a result, gender (defined as the proportion of women participants in a session) is included as a covariate in regression analyses that are compared to the convergence analysis results (Table B in S2 Text).”

Unfortunately, we did not collect information on other characteristics of the sample. We think this is an important avenue for future research. 

2) Seizure as a policy in the "real world" entails costs over and above the amount seized. As law enforcement detects illegal activity and the channels through which the profits are laundered, criminals must hire new agents or create new ways to launder their money. There is also a chance that they will lose all profits and exit the game entirely. I think this needs to be acknowledged as seizure of profits in the experiment simply sees the "criminals" lose tokens with some probability.

We agree that we did not highlight that in the real world, there are other costs beyond the loss of laundered profit. To address this issue, we have added the following to lines 343-345 on page 16:

“When law enforcement seizes laundered money, for example, criminals must find new avenues to launder their illicit profit, which can be very costly.” 

To note that we are modeling large entities in our market, and that smaller agents may lose all product or profit from law enforcement action, we have added the following to the Conclusion section on page 32 to lines 694-703: 

“Further, the buyers and sellers modeled in this study are assumed to represent large entities where law enforcement, even when successful, cannot seize all illegal assets at any stage along the production-trade-laundering cycle. It is thus assumed that operations modeled are large enough, for instance, that if one field of illegal crop is eradicated, there are many other fields left undetected, or illicit profit is laundered through multiple avenues, such that, even with detection of money laundering in some avenues, other avenues remain undetected. Yet, the potential exists for law enforcement to successfully seize all assets of smaller operations than assumed here. Additionally, our analysis does not capture the ability of operations to reinvest profits into capital acquisition to expand production capabilities or invest in other criminal activities. Future research should focus on how effective various law enforcement policies are in such situations.”

3) Are the participants equally distributed across the treatments?

Yes, participants were equally distributed across the treatments, i.e., the same number of participants were used for each experimental session, and different subjects were used for each session. We have added the following text to page 13, lines 274-276 of the main manuscript: 

“We use a between-subjects design where subjects were recruited to participate in only one session and treatment. Subjects selected a session to participate in, after which, the treatment was randomly assigned to each session. Each session had eight participants.”

Further, every attempt was made to randomize participants across the treatments. Yet, as stated prior, we do recognize that our attempts may still not lead to full randomization of participants across treatments. We do find that the sample is imbalanced in respect to participant gender and conduct analyses controlling for gender to ensure our results are not biased by such imbalance.

4) I am unfamiliar with the parametric convergence model and I think the rationale for using it should be discussed. In any event, for transparency and completeness I would like to see the analysis starting with a simpler method of comparing means across treatments and running regressions with treatment effects (including any personal characteristics that vary by treatment).

Thank you for your comment. We have added the following text to pages 21-22, lines 465-472 to more fully explain the rationalization for using the convergence analysis:

“The convergence model provides a statistical description of the path of the data from the beginning level to the asymptote of each treatment while addressing econometric issues found in the experimental data (such as heteroskedasticity, contemporaneous correlation, and serial correlation) [46]. As t increases, the weight of the starting value becomes smaller, while the weight of the asymptote becomes larger. Through this analysis, we can evaluate treatment effects by testing for differences in the estimated asymptotes for each treatment without making a priori judgements on the appropriate data that represents stable convergence, such as is needed in more simplistic difference of means tests or other tests that do not account for t.”

To add validity to the convergence analysis results, we have also included results from simultaneous difference of means tests (through Tukey tests) in Table A in S2 Text. Further since gender has been previously found to influence outcomes in such market experiments and is found to be imbalanced across the treatments, we also report regression analyses that include the percent of women in a session as a covariate in Table B in S2 Text. We have added the following to the main manuscript to explain the inclusion of the Tukey test and regression analyses to pages 22 and 23, lines 488-494: 

“To lend validity to the convergence analysis, results are compared to simultaneous difference of means two-tailed t-tests via Tukey tests and regression analyses over the last five periods (which is used in the previous literature to represent convergence [30-31,50]) when including covariates found to potentially influence market outcomes from previous literature (S2 Text). Although including covariates when measuring treatment effect leads to biased estimators [51], recent literature suggests that properly adjusting for any imbalanced prognostic variables is appropriate [52-53]. These additional analyses are performed via STATA statistical software.”

We find the results from the Tukey and regression analyses are consistent with the results from the convergence analysis. We have noted the consistency between the results in the manuscript. Specifically, on page 24, lines 520-523, we state: 

“These results are consistent with difference of means Tukey tests (Table A in S2 Text) and the regression analysis (the only difference to note is that the regression analysis found that the Both Profit Seizure treatment had significantly more units produced than the No Seizure treatment; Table C in S2 Text).”

On page 25 lines 541-543, we state: 

“Again, these results are consistent with the regression analysis and difference of means test (except that the Tukey test did not find a significant treatment effect for the Both Seizure treatment; Table A and C in S2 Text).”

On page 27, lines 578-580, we now state: 

“These results are supported by the regression and Tukey results, lending further support to these outcomes (Table A and C in S2 Text).”

We also note that results are, “consistent with regression and Tukey results; Table A and C in S2 Text” twice on page 27.

---

## [Decision Letter · Decision Letter 1]

23 Jun 2021

PONE-D-21-07819R1

The Relative Effectiveness of Law Enforcement Policies Aimed at Reducing Illegal Trade: Evidence from Laboratory Markets

PLOS ONE

Dear Dr. jones ritten,

Thank you for submitting your manuscript to PLOS ONE. After careful consideration, we feel that it has merit but does not fully meet PLOS ONE’s publication criteria as it currently stands. Therefore, we invite you to submit a revised version of the manuscript that addresses the points raised during the review process.

The two reviewers are divided: one of them recommends acceptance with minor revisions and the other recommends rejection. 

The positive reviewer sees a lot of merit and improvement in your revision of the paper but still calls for sharpening the message, especially in the introduction. You should take her/his criticism seriously and make a final careful revision of the passages that are mentioned in her/his report.

Reviewer #2 is more critcial and he has concerns about (i) randomization into treatments, (ii) balance, and (iii) statistical analysis. I agree with her/him that randomization is not perfect -- ideally each participant would have an i.i.d. chance of ending up in each treatment and this is something the experimental economics community should pay more attention to. You do not have many participant characteristics to judge your balance against but gender is suggestive of some, but maybe not all that dramatic, failure in this respect. You are very open about this challenge in the paper, so it is easy for the reader to discount when judging the evidence. 

Reviewer #2 is also critical about the statistical analysis and calls for panel regressions with data from all rounds. While I agree with that being a valid approach, I also see value in treating each session as an independent observation and looking at long run "equilibrium" effects. This approach has been used in the literature as you mention. The value of this approach is slightly downplayed by the fact that your theoretical predictions are suggestive at best. I would like you to add one more robustness check as I think the normality assumption in the Tukey tests could be a bit heroic: I'd like you to report Wilcoxon rank-sum tests (they can be one sided as you are clearly expecting a reduction in quantities etc.) session averages in the last five periods being the independent observations.

Let me emphasize that I cannot make any commitment to publish your work at this stage. I nevertheless look forward to receiving your revision soon enough. 

We look forward to receiving your revised manuscript.

Kind regards,

Topi Miettinen, PhD

Academic Editor

PLOS ONE

Journal Requirements:

Reviewers' comments:

Reviewer's Responses to Questions

**Comments to the Author**

1. If the authors have adequately addressed your comments raised in a previous round of review and you feel that this manuscript is now acceptable for publication, you may indicate that here to bypass the “Comments to the Author” section, enter your conflict of interest statement in the “Confidential to Editor” section, and submit your "Accept" recommendation.

Reviewer #1: All comments have been addressed

Reviewer #2: (No Response)

2. Is the manuscript technically sound, and do the data support the conclusions?

Reviewer #1: Yes

Reviewer #2: No

3. Has the statistical analysis been performed appropriately and rigorously? 

Reviewer #1: Yes

Reviewer #2: No

4. Have the authors made all data underlying the findings in their manuscript fully available?

Reviewer #1: Yes

Reviewer #2: Yes

5. Is the manuscript presented in an intelligible fashion and written in standard English?

Reviewer #1: Yes

Reviewer #2: Yes

6. Review Comments to the Author

Reviewer #1: The authors put a lot of effort in addressing my comments from the previous round of reviews, which is much appreciated.

The introduction is now clearer about the contribution of the paper.

However, I still believe the exposure can be improved:

In its current version, the introduction describes some empirical examples of seizure policies from lines 67 to 78. It then goes on about the recent interest on money laundering until line 112. Then the main argument of the paper is outline from line 113 to 123. This structure is not only a bit redundant, but also not straightforward.

A more natural flow would be to state the most important paragraph (from line 113 to 123) upfront. Then described the three policies while integrating the empirical examples (instead of keeping them separate, then making the point about money laundering policies and concluded on the fact that while policy makers focus on this one particularly, there is no evidence that it is where the seizure is most efficient. I would then keep the paragraph from line 124 to 135, as it is a nice transition to the rest of the paper.

Whether the above suggestions are implemented by the authors should not affect the final decision, as in my opinion the paper is acceptable in its current version. I am merely giving the authors one last opportunity to deliver the best possible version of their work.

I am globally satisfied with the proposed changes, and I believe the paper can be accepted for publication in its current version.

Reviewer #2: The authors have addressed many of my concerns but I still have reservations about the balance of the sample and the regression analysis. It is not clear to me why only 5 periods would be used (from the 20+ available). My understanding is that the model is estimated on the averaged values for those 5 rounds. Typically, these kinds of models would be estimated on the full panel data with period fixed effects to account for learning (and perhaps a dummy to capture if it were a "practice" round). Given how unbalanced the sample was in terms of gender, one worries that other unobserved/unmeasured characteristics could be driving the results. The fact that subjects "selected" a session to attend also speaks to the fear that different types of people ended up in different treatments in a non-random way. Ultimately, we just do not know enough about the composition of the sample to have confidence in the results.

7. PLOS authors have the option to publish the peer review history of their article (what does this mean?). If published, this will include your full peer review and any attached files.

Reviewer #1: No

Reviewer #2: No

---

## [Author Response · Author response to Decision Letter 1]

27 Jul 2021

Dear Dr. Miettinen,

Thank you for the opportunity to submit a second improved revision of the manuscript, “The Relative Effectiveness of Law Enforcement Policies Aimed at Reducing Illegal Trade: Evidence from Laboratory Markets.” We sincerely appreciate the additional comments provided by yourself and the two reviewers. We have attempted to address them all in the manuscript and in this response. We believe the manuscript is much improved because of these constructive comments. Below is a detailed response to the comments made by yourself and two reviewers, specifying changes to the original manuscript to address the concerns. Changes to the manuscript are noted by blue in the text.

Dr. Miettinen: Reviewer #2 is more critical and he has concerns about (i) randomization into treatments, (ii) balance, and (iii) statistical analysis. I agree with her/him that randomization is not perfect -- ideally each participant would have an i.i.d. chance of ending up in each treatment and this is something the experimental economics community should pay more attention to. You do not have many participant characteristics to judge your balance against but gender is suggestive of some, but maybe not all that dramatic, failure in this respect. You are very open about this challenge in the paper, so it is easy for the reader to discount when judging the evidence. 

Reviewer #2 is also critical about the statistical analysis and calls for panel regressions with data from all rounds. While I agree with that being a valid approach, I also see value in treating each session as an independent observation and looking at long run "equilibrium" effects. This approach has been used in the literature as you mention. The value of this approach is slightly downplayed by the fact that your theoretical predictions are suggestive at best. I would like you to add one more robustness check as I think the normality assumption in the Tukey tests could be a bit heroic: I'd like you to report Wilcoxon rank-sum tests (they can be one sided as you are clearly expecting a reduction in quantities etc.) session averages in the last five periods being the independent observations.

We agree that the normality assumption used by the Tukey tests may be heroic. We have replaced the Tukey tests with Wilcoxon rank-sum tests over the last five periods and report them in S2 Text. The results from the Wilcoxon rank-sum tests are consistent with the convergence and regression analyses. We reference the Wilcoxon’s non-parametric Rank-Sum tests throughout the revised manuscript. 

Further, in our response to Reviewer 2, we provide additional literature to support and clarify our approach and report further analyses to show our results are robust. Since our focus in the manuscript is on equilibrium outcomes, we only report these additional results in this response, and not in the manuscript itself (or supplemental appendices). 

Reviewer #1: The authors put a lot of effort in addressing my comments from the previous round of reviews, which is much appreciated.

Thank you

The introduction is now clearer about the contribution of the paper. However, I still believe the exposure can be improved:

In its current version, the introduction describes some empirical examples of seizure policies from lines 67 to 78. It then goes on about the recent interest on money laundering until line 112. Then the main argument of the paper is outline from line 113 to 123. This structure is not only a bit redundant, but also not straightforward.

A more natural flow would be to state the most important paragraph (from line 113 to 123) upfront. Then described the three policies while integrating the empirical examples (instead of keeping them separate, then making the point about money laundering policies and concluded on the fact that while policy makers focus on this one particularly, there is no evidence that it is where the seizure is most efficient. I would then keep the paragraph from line 124 to 135, as it is a nice transition to the rest of the paper.

Thank you for this very insightful comment. We have reorganized the introduction so that anti-money laundering policy is introduced earlier. This reorganization also integrates the empirical examples of the various policies, and then focuses on the current prevalence of anti-money laundering policies. Specifically, lines 66-111 on pages 3-5 now reads: 

“Although law enforcement polices exist that focus on the production or sale of illegal products, recent emphasis has been placed on heightened enforcement aimed at the laundering of illicit profit. Generally, illegal trading relies on cash transactions, where the cash must be disguised to avoid legal entanglements. This cash is frequently passed through banks and other financial institutions in order to make its source difficult to trace [6]. This form of money laundering disguises the source of proceeds from criminal activity by making it appear as if earnings are legitimate. The amount of cash flowing through illegal transactions is estimated to account for two to five percent of global GDP yearly [6]. In addition to potential societal harm from illegal trade activity, money laundering and the injection of laundered funds into new ventures create economic damage. It is estimated that an increase of USD 1 billion in money laundering can reduce economic growth between 0.03 and 0.06 percentage points [7]. 

Various other policies focus on other stages throughout the production-trade-laundering cycle. For instance, the Drug Enforcement Administration (DEA) uses crop eradication to reduce the quantity of illegal drugs before they can hit the market. In 2019, the DEA eradicated over 4 million marijuana plants and seized nearly $30 million dollars of cultivator assets [8]. Additionally, other policies include street-level enforcement to interrupt transactions and border enforcement [9] to capture assets during the sale of the illegal products. Border enforcement focuses on seizing illegal products as they cross international boundaries. Since most of these products are in transit to their buyer, seizure at the border, like street-level enforcement, aims law enforcement actions during the sale or trade of illegal products. Although these policies aimed at the other stages in the production-trade-laundering cycle can be impactful, anti-money laundering policies have been the focus of recent anti-crime campaigns.

Numerous national and international anti-money laundering policies and entities have been created over the last 40 years. Examples of such entities include the Financial Action Task Force, the Global Programme Against Money Laundering, and The International Convention Against Transnational Organized Crime, initiated through global organizations such as the United Nations and International Monetary Fund (see [10] for a review). Although increases in anti-money laundering enforcement from these programs is projected to decrease crime [11], many studies find that the effectiveness of these initiatives is limited (e.g. [12]). For instance, Ferwerda et al. [13] find that the Financial Action Task Force’s (an independent inter-governmental body that develops and promotes policies to protect the global financial system against money laundering) current method of blacklisting countries for money-laundering (used as an attempt to reduce money laundering) may not prevent money laundering, and may further reduce the quality of national statistics on money laundering. Further, Deleanu [12] finds evidence that current policy does not incentive accurate reporting by countries of money laundering, supporting the notion that national statistics on money-laundering activity are not accurate, and may even be misleading.

In another instance of how current anti-money laundering policies may have limited effectiveness, Takáts [14] finds that current polices requiring banking institutions to identify and report suspicion of money laundering may backfire. The Suspicious Activity Report, introduced in 1996 by the Financial Crime Enforcement Network for reporting of any suspicious activity, is one such policy. To incentivize compliance, banks and other financial institutions are charged a fee for not reporting instances of money-laundering. Current increases in this fee can cause institutions to over-report potential instances of money-laundering, creating a ‘cry wolf’ phenomenon, which reduces the efficacy of the program to reduce money-laundering, and thus crime [14].”

Whether the above suggestions are implemented by the authors should not affect the final decision, as in my opinion the paper is acceptable in its current version. I am merely giving the authors one last opportunity to deliver the best possible version of their work.

I am globally satisfied with the proposed changes, and I believe the paper can be accepted for publication in its current version.

Thank you

Reviewer #2: The authors have addressed many of my concerns but I still have reservations about the balance of the sample and the regression analysis. It is not clear to me why only 5 periods would be used (from the 20+ available). My understanding is that the model is estimated on the averaged values for those 5 rounds. 

Thank you for your comments. The convergence regression is based on all of the market data for 20 periods (to have balanced panel data) for each treatment, not just five periods (except for the variable, Units Produced, since it is not normally distributed, and a nonparametric analysis is conducted). The model does put less weight on earlier periods than later periods to account for learning. Thank you for highlighting this source of confusion. We have now added the following text to the footnote for Table 3 on lines 519-522:

“Units Produced do not meet normality and are severely skewed as per Brown [49]. Thus, following previous research, we report averages for the last 5 trading periods per treatment and non-parametric test results [30-31,50]. All other variables meet the normality assumption and the convergence results and parametric tests are reported using data from all 20 trading periods”.

Using the later trading periods for testing treatment effects (for variables that are not normally distributed) is a standard practice in the literature for market experiments focusing on equilibrium outcomes (e.g., Sabasi et al., 2013, Phillips et al., 2003, Davis and Holt, 1993, Mestelman and Welland, 1988). 

We only use the last five periods in the supplemental analyses since variability from learning across subjects and treatments could impact the ability of parameter estimates to capture the magnitude of treatment effects after learning. 

Noussair CN, Plott CR, Riezman RG. 1995. “An experimental investigation of the patterns of international trade.” American Economic Review 85(3):462-91.

Mestelman, S., and D. Welland. 1988. Advance production in experimental markets.” Review of Economic Studies. 55,4: 641-654.

Davis, D. D. and C. A. Holt. 1993. “Chapter 9 Economic Behavior and Experimental Methods: Summary and Extensions.” In Experimental Economics. Princeton University Press, Princeton, NJ.

Phillips, O. R., D. J. Menkhaus, and K. T. Coatney. 2003. “Collusive Practices in Repeated English Auctions: Experimental Evidence on Bidding Rings,” American Economic Review, 93,3: 965-979.

Sabasi, D. M., C. T. Bastian, D. J. Menkhaus, and O. R. Phillips. 2013. “Committed Procurement in Privately Negotiated Markets: Evidence from Laboratory Markets,” American Journal of Agricultural Economics, 95,5:1122-1135.

Typically, these kinds of models would be estimated on the full panel data with period fixed effects to account for learning (and perhaps a dummy to capture if it were a "practice" round). 

We do agree that various econometric approaches are used in the literature. Yet, in laboratory market studies focusing on equilibrium outcomes (not individual outcomes), the convergence analysis is an established technique (see for example Plott and Pogorelskiy, 2017, and the other references noted in the manuscript). The benefit of using the convergence model regression method (Noussair et al. 1995) instead of fixed effects regression analysis, is that the convergence analysis addresses all the panel data characteristics that could impact econometric efficiency issues. Further, the convergence analysis more accurately follows the path of the data to test for differences in asymptotes, which are the statistically predicted equilibrium outcomes from the data for each experimental treatment.

We have conducted a period fixed effects regression below (using the full panel data) to provide further validity to the results from the convergence analysis. We do find more statistically significant results than from either the convergence analysis or the regression analysis only including the last five periods. We note differences in yellow (see Table below). These differences only further highlight our general conclusion that anti-money laundering policies are likely ineffective at reducing crime (unlike in the convergence and regression analyses on only the last five periods, we find a significant increase in trades in the Seller Profit Seizure treatment when compared to the No Seizure treatment: see Table below). The results also further support our other key finding that only when law enforcement is aimed at trade is crime likely reduced (only in the Trade Seizure treatment are both Units Produced and Units Traded significantly lower than in the No Seizure treatment). 

 Dependent Variable

Independent Variable 

(No Seizure is the Omitted treatment) Units Produced Units Traded Price Seller Earnings Buyer Earnings Total Earnings

Seller Profit Seizure 0.437 0.813* -2.917* -19.92* 7.647* -49.38*

Buyer Profit Seizure -0.139 -0.115 -1.709 -6.401 -25.26* -126.80*

Both Profit Seizure 1.518* 1.917* 2.550* -2.11 -30.87* -132.10*

Product Seizure 1.268* -1.59* 4.856* 32.71* -28.70* -245.80*

Trade Seizure -1.952* -1.650* -2.463* -58.88* -42.68* -406.4*

Percent Women (in Session) 2.690* 3.208* 12.07* 52.13* -8.39 191.2*

Constant 14.19* 13.05* 68.48 68.71* 149.00* 870.90*

F-stat 20.09 35.06 46.45 35.06 46.45 120.54

We do not report these results in the manuscript since the model’s inability to account for characteristics that could impact econometric efficiency issues potentially suggests statistical differences when none truly exist. Moreover, this period fixed effects model could be underspecified and have biased parameter estimates. Thus, to limit the risk of Type I errors, the convergence analysis is reported in the manuscript.

We also have conducted an alternative Time Series Cross Sectional Model which includes treatment dummies, a trend, and a fixed effects dummy for learning using all 20 periods of data and estimated with the PARKS method panel data estimator in SAS (see Table below). This model is more appropriate than the fixed effects regression since it accounts for some of the characteristics in the data that can affect econometric efficiency. It also is better specified in that it addresses potential trends in the data over time. The results of this model are also similar to the convergence model results, and again lends further support to the findings reported in the manuscript. The R-squares are lower in this model since it does not follow the path of the data (i.e., predict the dependent variable as well) and our estimates of converged values at equilibrium from this model are different than the convergence model in several instances given the concerns previously expressed regarding this approach. 

 Dependent Variable

Independent Variable 

 Units Produced Units Traded Price Seller Earnings Buyer Earnings Total Earnings

Seller Profit Seizure 0.312 0.867* -4.194* -22.861* 8.812 -62.729*

Buyer Profit Seizure -0.328 0.087 -4.278* -13.589 -25.040* -155.836*

Both Profit Seizure 1.204* 2.068* 0.126 -8.116 -30.032* -158.433*

Product Seizure 1.147* -1.422* 3.634* -36.579* -28.451* -258.920*

Trade Seizure -1.905* -4.680* -2.111* -57.771* -42.804* -402.138*

Percent Women (in Session) 0.064 0.418* -0.32 0.933 -0.548 1.724

Learn Dummy 0.535* 0.114 1.54 -1.634 0.118 -4.181

Trend -0.010 -0.234* 0.497* 1.054 -0.949 0.708

Constant 15.956* 14.875* 70.134* 79.436* 159.542* 952.074*

R-Square 0.8608 0.8492 0.8441 0.8342 0.9133 0.9125

These results also support the conclusion in the manuscript that the amount of illegal product available and traded in the market is reduced most when seizing units during trade. Further, the conclusion that illicit trade is either increased or not statistically different from the base treatment when enforcement is aimed at money laundering is again consistent with the results presented in the manuscript. 

The practice rounds used different unit costs for sellers and different redemption values for buyers (which is common practice to not have participants establish expectations for the actual experiment). Any regression analyses using the practice rounds would be unfitting as they would have different predicted equilibrium values than the actual data. 

Plott, C. R., and K. Pogorelskiy. 2017. “Call Market Experiments: Efficiency and Price Discovery through Multiple and Emergent Newton Adjustments,” American Economic Journal: Microeconomics 9,4: 1-41.

Given how unbalanced the sample was in terms of gender, one worries that other unobserved/unmeasured characteristics could be driving the results. 

The market experiments used in this study rely on induced value theory (Smith, 1976) where “individuals’ innate characteristics become largely irrelevant” (Friedman and Sunder, 1994, p.13). Moreover, as we are interested in market equilibrium effects of policy, we run six replications to further reduce the potential impact of individual impacts. Since we use multiple trading periods, with three bargaining rounds each, across multiple replications, and random assignment of treatment and agent role, the expectation is that subject pool/sample effects are generally reduced or washed away. 

A possible solution to address the potential that unobserved characteristics are driving the results is to have used a different sample choice. As we do not know the population characteristics of those involved in illegal product markets, we cannot draw a sample representative of the actual population involved in such markets. 

Further, Frechette (2015) reviews the experimental literature to investigate the use of representative or demographically varied samples, and finds, “As a whole, there are a few patterns that seem to emerge from the use of demographically varied samples. One is that results are often not drastically different from those using the standard sample of student subjects, certainly when it comes to comparative statics, and to the extent that there are differences, these can often be traced to age (over 55)” (p. 460). We do not expect our sample to vary much by age because the community where the experiment was conducted has a very young average age given the presence of the University. 

Smith, V. L. (1976). “Experimental Economics: Induced Value Theory,” American Economic Review. 66: 274-279. 

Friedman, D. and S. Sunder. 1994. “Chapter 2 Principles of Economic Experiments.” In Experimental Methods: A Primer for Economists. Cambridge University Press, New York, NY.

Fréchette GR. Experimental economics across subject populations. In: Kagel JH, Roth AE, editors. The handbook of experimental economics, volume 2. Princeton: Princeton University Press; 2015, pp 435-480.

The fact that subjects "selected" a session to attend also speaks to the fear that different types of people ended up in different treatments in a non-random way. 

Participants did sign up for a time as you suggest, but the treatment to be run was chosen without the participant’s knowledge. Further, participants were randomly chosen by the computer to be either a buyer or seller in that session after logging in. 

Ultimately, we just do not know enough about the composition of the sample to have confidence in the results.

We hope these comments and further analyses address your concerns regarding our analyses, results, and conclusions as it relates to our sample of participants.

---

## [Editor Report · Decision Letter 2]

9 Aug 2021

PONE-D-21-07819R2

The Relative Effectiveness of Law Enforcement Policies Aimed at Reducing Illegal Trade: Evidence from Laboratory Markets

PLOS ONE

Dear Dr. jones ritten,

Thank you for submitting your manuscript to PLOS ONE. After careful consideration, we feel that it has merit but does not fully meet PLOS ONE’s publication criteria as it currently stands. Therefore, we invite you to submit a revised version of the manuscript that addresses the points raised during the review process.

You have done an excellent job with the revision and I am pleased to inform you that we are almost there. You should consider this decision as conditional acceptance. There are three final edits, I'd like you to address. 

1. That seller profits fall short of buyer profits is indicative of potential implications regarding the incidence of deterrence, simlar to that in the case of tax evasion. Are you in a position of saying something interesting about that and relating to existing literature? It seems in each case, you might be able to calculate the expected direct  monetary consequences of seizure on sellers and then calculate to which extent the sellers can pass the cost onto buyers as higher prices. This can be very brief if you prefer, some summary of data on suggestive of a future line of inquiry. I may be mistaken here, in which case explain in your response why you are unable to do this.   

2. The following recent publication seems related in that it attempts to address the question where deterrence is most effective. Are there other similar contributions which you could briefly relate to? Banerjee, R., Boly, A., & Gillanders, R. (2020). Anti-Tax Evasion, Anti-Corruption and Public Good Provision: An Experimental Analysis of Policy Spillovers. *Available at SSRN*.

3. Please, check your list of references carefully. For instance, Fismen & Miguel -> Fisman & Miguel

We look forward to receiving your revised manuscript.

Kind regards,

Topi Miettinen, PhD

Academic Editor

PLOS ONE
---

## [Author Response · Author response to Decision Letter 2]

13 Sep 2021

Dear Dr. Miettinen,

Thank you for the opportunity to resubmit the manuscript titled, “Money Laundering, Crime, and the Seizure of Assets: Relative Effectiveness of Law Enforcement Policies in Experimental Bargaining Markets,” to address your comments. Our response to each of your excellent comments are detailed below:

1. That seller profits fall short of buyer profits is indicative of potential implications regarding the incidence of deterrence, simlar to that in the case of tax evasion. Are you in a position of saying something interesting about that and relating to existing literature? It seems in each case, you might be able to calculate the expected direct monetary consequences of seizure on sellers and then calculate to which extent the sellers can pass the cost onto buyers as higher prices. This can be very brief if you prefer, some summary of data on suggestive of a future line of inquiry. I may be mistaken here, in which case explain in your response why you are unable to do this. 

Thank you for this useful comment. We now include a table and discussion on the incidence of deterrence to both buyers and sellers on Lines 600-642:

The increased bargaining power of buyers can be further seen by the amount of the cost of enforcement absorbed by sellers compared to buyers. Based on the theoretical predications presented in Fig 1 and Table 2, the incidence of enforcement (the percent decrease in Total Earnings from enforcement absorbed by the relevant party) for both sellers and buyers are shown in Table 5. Based on theory, sellers should only have a higher enforcement incidence than buyers when enforcement is aimed at money laundering of sellers only (Table 5). Yet, the experimental results show that sellers additionally absorb more enforcement cost than buyers in the Product Seizure and Trade Seizure treatments (the percent decrease in Total Earnings absorbed by the relevant group from Table 4 and shown in Table 5). Further, comparing the predicted seller enforcement incidence to the realized (from the experimental data) seller enforcement incidence in Table 5, sellers have a higher enforcement incidence than expected in the Buyer Seizure, Product Seizure, and Trade Seizure treatments. Thus, buyers are able to use their increased bargaining power in these treatments to push some of their cost of enforcement onto sellers.

Table 5: Enforcement Incidence by Treatment (in Percent)

 Treatment 

Variable Seller Profit Seizure Buyer Profit Seizure Both Profit Seizure Product Seizure Trade Seizure

Predicted Seller Enforcement Incidence i 100 0 50 28 26

Predicted Buyer Enforcement Incidence ii 0 100 0 72 74

Realized Seller Enforcement Incidence iii 92 20 21 54 56

Realized Buyer Enforcement Incidence iv 8 80 79 46 44

Difference between Expected and Realized Incidence v 8 -20 29 -26 -30

 i Calculated as the expected loss in Seller Earning, divided by the expected loss in Total Earning (Seller and Buyer Earnings) in the given treatment compared to the No Seizure treatment. Values are based on the theoretical predictions presented in Table 2.

ii Calculated as the expected loss in Buyer Earning, divided by the expected loss in Total Earning (Seller and Buyer Earnings) in the given treatment compared to the No Seizure treatment. Values are based on the theoretical predictions presented in Table 2.

iii Calculated as the realized loss in Seller Earning, divided by the realized loss in Total Earning (Seller and Buyer Earnings) in the given treatment compared to the No Seizure treatment. Values are based on the convergence analysis presented in Table 4.

iv Calculated as the realized loss in Buyer Earning, divided by the realized loss in Total Earning (Seller and Buyer Earnings) in the given treatment compared to the No Seizure treatment. Values are based on the convergence analysis presented in Table 4.

v Calculated as the difference between expected and realized incidence for sellers for the given treatment. A positive value indicates a reduction in realized incidence, compared to theoretical predictions, for sellers (a benefit to sellers). A negative value indicates a reduction in realized incidence, compared to theoretical predictions, for buyers (a benefit to buyers).

It is only in the Seller Profit Seizure and Both Profit Seizure treatments that sellers are able to pass on some of their expected loss in earnings to buyers (Table 5). But in neither of these treatments were sellers able to pass on enough of the cost of enforcement for sellers to have higher earnings than buyers (Table 4). As such, buyers seem to have a bargaining advantage over sellers in illegal product markets.

 Table 5 also shows that theoretical predications of incidence, which depend solely on the relative elasticity of supply and demand [57], may fail to hold for enforcement incidence in bilateral bargaining markets. This result is similar to other experimental studies of incidence (e.g. [33]), especially in markets with bilateral bargaining [31-32], since trading institution has been shown to impact incidence [58]. 

We have also added the following references:

Ruffle BJ. Tax and subsidy incidence equivalence theories: experimental evidence from competitive markets. J Public Econ 2005;89:1519-42.

Cox JC, Rider M, Sen A. Tax incidence: Do institutions matter? An experimental study. Public Finance Rev 2017;46(6):899-925. 

Unfortunately, due to the structure of our experiment, we do not feel we can relate our results directly to the tax evasion literature. Many of these experiments, such as those you have conducted, allow participants to make a decision to evade taxes. In real world illegal markets, participants may expend efforts to not get caught or perhaps find themselves in a position to bargain for others to get caught, such as in a prisoner’s dilemma situation. However, as the probability of seizure is exogenous in our experiment, we do not have this element of choice in our experiment. Thus, the incentives placed on participants, we feel, may cause different behavior in our study compared to the tax evasion literature. Yet, we feel this is an important element, and hope it will lead to future work in this area. 

2. The following recent publication seems related in that it attempts to address the question where deterrence is most effective. Are there other similar contributions which you could briefly relate to? Banerjee, R., Boly, A., & Gillanders, R. (2020). Anti-Tax Evasion, Anti-Corruption and Public Good Provision: An Experimental Analysis of Policy Spillovers. Available at SSRN.

Again, thank you for this note. We now include the following to Lines 583-587 to state that our findings are similar to studies on enforcement on other types of crimes that find a spillover effect:

When law enforcement seizes products during trade, there may be a spillover effect with sellers potentially leaving the market in the long run, which may further reduce illegal market activity and crime. This result is similar to other experimental studies that show effective enforcement focused on other sources of crime (e.g. tax evasion) have spillover effects that may further help in reducing crime [54-56]. 

We have also added the following references:

Banerjee R, Boly A, Gillanders R. Anti-tax evasion, anti-corruption and public good provision: An experimental analysis of policy spillovers. SSRN. 2020; Available from: http://dx.doi.org/10.2139/ssrn.3652411.

Boly A, Gillanders R, Miettinen T. Deterrence, contagion, and legitimacy in anticorruption policy making: an experimental analysis. J Leg Stud 2019;48:275-305.

Braga AA, Apel RJ, Welsh BC. The spillover effects of focused deterrence on gang violence. Eval Rev 2013;37:314–342. 

3. Please, check your list of references carefully. For instance, Fismen & Miguel -> Fisman & Miguel

Thank you for pointing out the errors in our references. We have corrected these errors.

---

## [Editor Report · Decision Letter 3]

18 Oct 2021

The Relative Effectiveness of Law Enforcement Policies Aimed at Reducing Illegal Trade: Evidence from Laboratory Markets

PONE-D-21-07819R3

Dear Dr. jones ritten,

We’re pleased to inform you that your manuscript has been judged scientifically suitable for publication and will be formally accepted for publication once it meets all outstanding technical requirements.

I spotted one final typo (I think) you should correct. Please be in touch with the journal office and refer to this mail if you agree with me. On lines 633 and 634 you say: "It is only in the Seller Profit Seizure and Both Profit Seizure treatments that sellers are 634 able to pass on some of their expected loss in earnings to buyers (Table 5)." But should not this read: "It is only in the Buyer Profit Seizure and Both Profit Seizure treatments that sellers are 634 able to pass on some of their expected loss in earnings to buyers (Table 5)."

Kind regards,

Topi Miettinen, PhD

Academic Editor

PLOS ONE

Additional Editor Comments (optional):

I spotted one final typo (I think) you should correct. Please be in touch with the journal office and refer to this mail if you agree with me. On lines 633 and 634 you say: "It is only in the Seller Profit Seizure and Both Profit Seizure treatments that sellers are 634 able to pass on some of their expected loss in earnings to buyers (Table 5)." But should not this read: "It is only in the Buyer Profit Seizure and Both Profit Seizure treatments that sellers are 634 able to pass on some of their expected loss in earnings to buyers (Table 5)."
---

## [Editor Report · Acceptance letter]

25 Oct 2021

PONE-D-21-07819R3 

The Relative Effectiveness of Law Enforcement Policies Aimed at Reducing Illegal Trade: Evidence from Laboratory Markets 

Dear Dr. Jones Ritten:

I'm pleased to inform you that your manuscript has been deemed suitable for publication in PLOS ONE. Congratulations! Your manuscript is now with our production department. 

Kind regards, 

on behalf of

Prof. Topi Miettinen 

Academic Editor

PLOS ONE